# Using machine learning to simultaneously quantify multiple cognitive components of episodic memory

Soroush Mirjalili ⬡ ✉ & Audrey Duarte

Why do we remember some events but forget others? Previous studies attempting to decode successful vs. unsuccessful brain states to investigate this question have met with limited success, potentially due, in part, to assessing episodic memory as a unidimensional process, despite evidence that multiple domains contribute to episodic encoding. Using a machine learning algorithm known as "transfer learning", we leveraged visual perception, sustained attention, and selective attention brain states to better predict episodic memory performance from trial-to-trial encoding electroencephalography (EEG) activity. We found that this multidimensional treatment of memory decoding improved prediction performance compared to traditional, uni-dimensional, methods, with each cognitive domain explaining unique variance in decoding of successful encoding-related neural activity. Importantly, this approach could be applied to cognitive domains outside of memory. Overall, this study provides critical insight into the underlying reasons why some events are remembered while others are not.

A major unanswered question in psychology is why do we remember some events yet forget others? Understanding this question is important not only for basic science but also for potential interventions that might improve learning in real time in a variety of populations and real-world settings such as the workplace or classroom. Recent advances in machine learning have made it possible for cognitive neuroscientists to explore the "brain states" occurring during learning that predict successful episodic memory for individual events, yet our ability to effectively predict memory performance from neural activity remains weak[1–4]. Thus, the likelihood that a reliable and practical memory intervention system could soon be realized is low.

One of the potential reasons for our inability to reliably predict memory performance from encoding-related brain activity is that previous studies have investigated episodic memory as a unidimensional process. Specifically, when training their memory decoders using cross-validation, those studies only considered the outcome of each event (i.e., whether it was later remembered or not) while overlooking the underlying processes that may underly episodic encoding. Critically, episodic memory is believed to be a multidimensional process in which various cognitive functions including perception[5,6],

sustained attention[7,8], selective attention[9,10], etc. contribute to memory formation. For example, neuroimaging evidence has shown some common brain areas engaged by multiple cognitive tasks, with activity levels supporting performance in tasks assessing episodic memory and other cognitive functions such as sustained attention and perception[6,7]. Evidence showing that direct current stimulation of brain areas in the selective attention network during episodic memory encoding[10] and presenting to-be-encoded stimuli during high vs. low sustained attention brain states[8] enhance episodic memory performance further support the multidimensionality of episodic memory. However, despite episodic memory's multidimensional nature, no study has considered the simultaneous involvement of multiple cognitive functions during episodic encoding to better understand the underlying reasons why any specific event was not successfully encoded. For example, was an event not encoded because the individual did not sufficiently perceive it? Did they fail to maintain their attention to successfully encode the event? Were they not selectively attending to that event and effectively ignoring distracting information? Or was encoding unsuccessful due to the failure to sufficiently engage multiple of these processes?

Department of Psychology, University of Texas at Austin, Austin, TX 78712, USA. ✉e-mail: soroushmirjalili@utexas.edu

Importantly, memory task performance is the outcome of the engagement of several cognitive processes but this binary outcome (i.e., remembered/forgotten) does not allow us to disentangle the contribution of these the associated underlying processes. As such, if we could collect neural activity from external tasks that were each associated with one of these underlying cognitive processes, we could covertly monitor their contribution to encoding-related brain activity. To elaborate with an example, one can record neural activity as participants perform a perception task and train a high vs. low perception performance classifier. This high vs. low performance classifier provides critical information regarding the features that best distinguish high and low perception brain states. Using a machine learning algorithm known as "transfer learning"[11–16], one can then leverage the information about low and high levels of perception and transfer the gained knowledge to encoding-related brain activity to predict how high or low the subject's perception was for each to-be-encoded event. Using this approach to break down episodic memory into its supporting cognitive components, we can better understand the extent to which they are engaged during episodic encoding. Specifically, transfer learning allows us to leverage information from external tasks (i.e., the "sources") that each engages a specific cognitive function (e.g., perception, sustained attention, etc.) essential for memory encoding (i.e., the "target"). Notably, only a handful of EEG studies have used transfer learning with multiple sources with the sources typically being EEG activity associated with the same cognitive task but from different participants with the aim of designing a subject-independent classifier that could potentially be useful for a brain-computer interface (BCI) system[15,17–20]. However, the current study uses this multidimensional approach to tease apart a cognitive process into its different cognitive components within participants. Although there are several cognitive functions linked to episodic encoding, considering all of them in this study was not practical due to the length of the experiment session. We used visual perception, sustained attention, and selective attention as the sources as they have been closely linked to episodic encoding[5–10] and their associated tasks are simple while allowing us to perform a high vs. low performance classifier. We hypothesize that this multidimensional evaluation of episodic encoding will allow us to better predict memory outcomes and more importantly, better understand why some events are remembered and others are not.

In order to better define brain states associated with memory success and failure, there are additional factors that need to be considered. Specifically, it may also be important to know how long a participant has been encoding events (i.e., the "time-on-task" effect), and whether the previously presented event was successfully encoded (i.e., the encoding "history"). However, previous studies attempting to distinguish successful from unsuccessful encoding overlooked these trial-to-trial variation within trial types. Regarding the time-on-task effect, it has been suggested that the neural resources for effective encoding cannot be sustained indefinitely, with resources becoming depleted (potentially due to fatigue) after a long period of continuous encoding[21–23]. Intracranial EEG[21] and fMRI[22] evidence has shown that higher time-on-task during episodic encoding is associated with diminished activity in brain regions that support episodic encoding success including the hippocampus, precuneus, and posterior cingulate. Similarly, other imaging studies have shown that with more time spent performing perceptual learning, sequential learning, and object naming tasks, activity in the associated brain regions is reduced[24–26]. We hypothesize that the neural evidence for high levels of each source will decrease as a function of the time that an individual has been encoding events. Additionally, as events are embedded within a temporal context that is sustained beyond a single episode[27,28], and as episodic encoding and retrieval processes evoke lingering states[29,30], brain states associated with a particular event will likely retain the "history" of prior brain states. Using transfer learning, we can determine whether the extent to which one or more cognitive functions is engaged reflects the history of the engagement of those functions for recently encoded events, further elucidating what underlies successful memory encoding. Specifically, we hypothesize that an event is more likely to involve higher levels of the underlying cognitive processes when it is preceded and followed by higher levels of engagement from those processes.

In this study, we test the idea that by investigating the trial-to-trial fluctuations in the levels of engagement of sustained attention, selective attention, and visual perception processes during episodic encoding, we can improve our ability to successfully predict, from encoding-related brain activity, which events will be later remembered vs. forgotten and the underlying reasons why. We recorded electroencephalography (EEG) while 43 young adults performed visual perception, sustained attention, selective attention, and episodic memory tasks. We designed a high vs. low performance classifier for each of the attention and perception tasks' EEG data and used transfer learning to leverage the information about the brain states associated with these low and high levels to predict episodic memory performance from trial-to-trial encoding EEG activity (Fig. 1)[20,31,32].

## Results

### Investigating episodic memory as a multidimensional process improved the memory prediction accuracy

Critically, we found that transfer learning significantly enhanced our ability to predict episodic memory success across participants compared to the traditional/unidimensional approach (from 72.0% to 81.4%) [$t(42) = 11.046, p < 0.001$, one $-$ tailed, $d = 0.81$; Fig. 2]. Two control analyses were performed to verify the validity of this multidimensional approach. First, one could argue that the reason this multidimensional transfer learning approach improved prediction performance is simply due to adding any information from any external source. Importantly, the transfer learning algorithm operates under the assumption that the high levels of the sources map on the high levels of the target (i.e., hits) and low levels of the sources map on the low levels of the target (i.e., misses) (Supplementary Fig. 8). If this assumption was not correct, the transfer learning would perform just as well if the source labels were randomly assigned, indicating the irrelevance of the brain states associated with the sources when predicting encoding success (more detail in the Supplementary information, "Control analyses for validating the transfer learning results"). However, we found that the classification performance was significantly worse than when the source labels were not shuffled [$t(42) = 6.15, p < 0.001$, one $-$ tailed, $d = 0.47$; Fig. 2], suggesting high levels of the sources do contribute to successful encoding encoding. Interestingly, this approach performed higher than the unidimensional approach [$t(42) = 3.80, p < 0.001$, one $-$ tailed, $d = 0.31$; Fig. 2]. One explanation is that some of these random assignments across the three sources lead to correlated labels with the original version and some sort of positive transferring will occur. Another possibility is that, in terms of associated neural activity, high and low trials from a given source are still quite similar to each other (i.e., fast and slow attention trials) and adding the information from the three sources will still benefit the encoding prediction even though the transfer is remarkably less optimal.

Additionally, it is important to note that the training portion of the encoding data is available for transfer learning to make necessary adjustments when transferring a source to the target. To elaborate, EEG data for each source task is associated with features that best distinguish high and low levels of the associated cognitive function (e.g., selective attention) on a subject-by-subject basis. However, the decision boundary to determine high vs. low levels of each underlying cognitive function during encoding needs to be adjusted according to range of feature values for hits and misses (Supplementary Fig. 8; see Methods for more detail). In this regard, we performed a second control analysis to investigate how much memory classification performance would be affected if

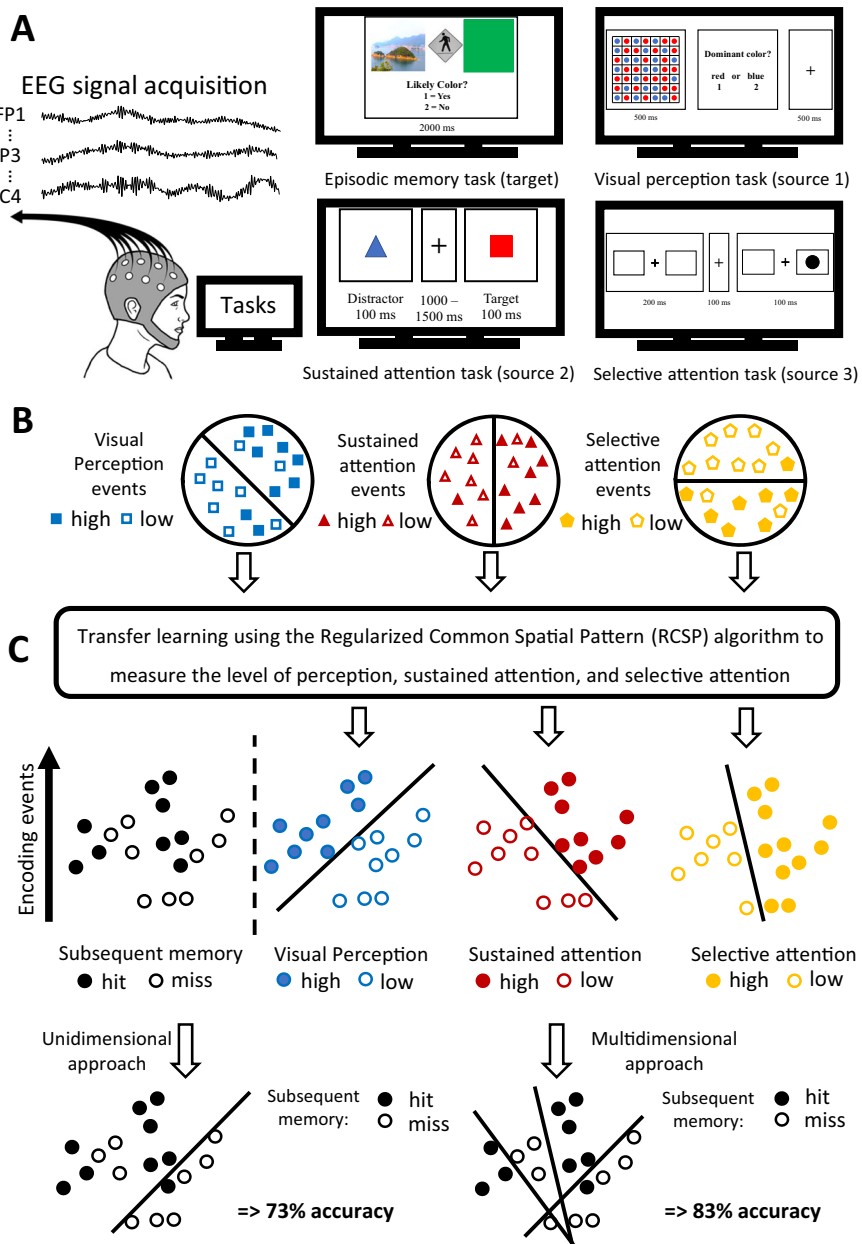

**Fig. 1 | General procedure. A** We collected EEG while the participants performed an episodic memory task (the target, more detailed illustration in Supplementary Fig. 4) as well as visual perception, sustained attention, and selective attention tasks (the sources). We used Illustrator to create the picture for the person wearing the EEG headset. The island in the episodic memory task is taken from Creative Commons. The image is available under the following Creative Commons license: https://creativecommons.org/licenses/by/4.0/. **B** For each source task, using the associated EEG for each participant, we trained a high vs. low performance classifier. Each colored shape represents a trial for each separate task. **C** After finding what features best distinguish high from low levels of performance for each source, we used the Regularized Common Spatial Pattern (RCSP) algorithm to predict whether perception, sustained attention, and selective attention are high or low during encoding events. The circles represent encoding events. We predicted that this multidimensional assessment of the underlying processes happening during encoding would improve memory decoding performance relative to evaluating episodic encoding as a unidimensional process (i.e., training the classifier with a portion of the encoding data and testing it on the remaining portion of the encoding data using cross-validation). The drawn decision boundaries and associated performance levels are shown for demonstrative purposes, not reflective of actual results.

there was no training data available from the encoding task and therefore, the sources were transferred to the encoding-related activity without making any adjustments. We found that the memory prediction performance significantly decreased compared to when the necessary adjustments were made (i.e., the training portion of the encoding data was available) during transfer learning [$t(42) = 15.18, p < 0.001$, one − tailed, $d = 1.05$] and compared to the unidimensional approach [$t(42) = 1.90, p = 0.032$, one − tailed, $d = 0.16$] (Fig. 2). These findings confirm the importance of using transfer learning to make necessary

adjustments (using the training portion of the encoding data) when transferring a source to the target[11–16].

## Every cognitive function explained additional unique variance of encoding-related activity

While these prior analyses showed that including all three sources significantly improved memory prediction performance compared to only using the training data from the memory task (i.e., unidimensional approach), they did not show how much each source individually

## Unidimensional vs. multidimensional classification

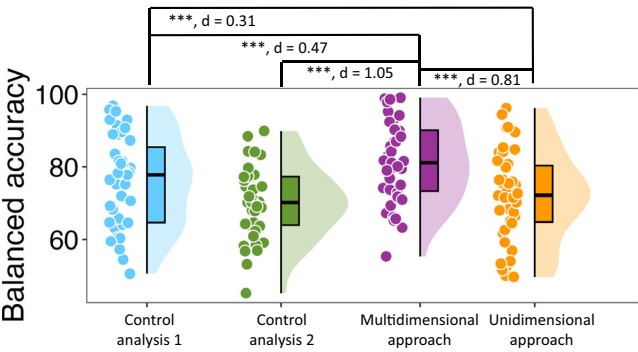

**Fig. 2 | Comparing the unidimensional and multidimensional classification approaches.** Comparisons of balanced accuracy for classifying encoding events (based on item memory success) using only the memory training data (i.e., the unidimensional approach) with transfer learning leveraging information from external sources to make a prediction. The results of two control analyses are also shown. The first control analysis tests the importance of using meaningful sources predicted to support memory encoding (i.e., perception, sustained attention, selective attention) to improve classification performance. We randomly assigned high/low labels to the source trials and repeated the transfer learning process; more detail in Supplementary Information). Classification performance was significantly lower than when the source labels were not shuffled [$t(42) = 6.15, p < 0.001$, one − tailed, $d = 0.47$], suggesting the relevance of the selected sources to encoding success. The second control analysis tests the importance of including the training portion of the memory encoding EEG data to allow transfer learning to make essential adjustments to effectively transfer each source to the memory data. It shows not making any adjustments (because of not having any training portion available from the memory data) can impair the transfer learning performance [$t(42) = 15.18, p < 0.001$, one − tailed, $d = 1.05$]. Circles reflect the data points of individual participants ($N = 43$). In the box plots, the minima represent the lowest data point within a condition, maxima represent the highest data point, centre represent the median value within the box, bounds of the box are the 25th and 75th percentiles, whiskers extend from the box to the minimum and maximum values that are not considered outliers, and percentile refers to the position of a data point within the distribution, with the box representing the middle 50% of data points between the 25th and 75th percentiles. The asterisks reflect statistically significant differences using one-tailed tests across conditions using Holm-Bonferroni corrections for multiple comparisons (***$p < 0.001$) and the associated Cohen's d is shown for each comparison. Source data are provided as a Source Data file.

contributed to this improvement. Specifically, is one of visual perception, sustained attention, and selective attention more important than the other two cognitive functions at different encoding periods for successful memory formation? As seen in Supplementary Fig. 10C, across all participants, there was no cognitive function at a particular encoding period that was consistently more important than other cognitive functions at other encoding periods (more detail in the Supplementary information, "Importance of different cognitive functions throughout the encoding period"). There was, however, a great amount of subject-to-subject variation for the extent to which each source contributed to encoding success at different encoding periods (Supplementary Fig. 10A, 10B). In addition, do visual perception, sustained attention, and selective attention explain unique variance of encoding-related activity? And does adding the second and third source improve memory prediction performance to the same extent that adding the first source? To answer these questions, we determined how much memory classification performance would improve by adding each source in a stepwise fashion. We added the sources in all six possible orders (e.g., 1st visual perception, 2nd sustained attention, 3rd selective attention). The patterns of results were similar across these orderings and thus, we report the average findings. There was a 4.9% performance improvement after the first source was added, regardless of order.

There was a 2.7% performance improvement once the second source was added, followed by a 1.8% performance improvement once the third source was added (Fig. 3). Statistical comparisons confirmed that adding each additional source significantly improved memory classification performance [all $ts > 5.968$, all $ps < 0.001$, one − tailed, all $ds > 0.16$]. Moreover, the extent to which classification performance increased by adding a source decreased with each step [step 2 improvement compared to step 1: $t(42) = 2.751, p = 0.004$, one − tailed, $d = 0.63$ and step 3 improvement compared to step 2: $t(42) = 1.080, p = 0.143$, one − tailed, $d = 0.27$].

### Time-on-task significantly impacted the engagement of perception, sustained attention, and selective attention during encoding

We then investigated whether the level of engagement of perception, sustained attention, and selective attention fluctuated depending on how long the participant had been performing the encoding task. We computed the slope of evidence values against the number of trials encoded across subjects. Consistent with our hypothesis, we found that as the time-on-task increased, the level of perception, sustained, and selective attention significantly decreased for both hits and misses [all $\rho s < −0.592$, all $ps < 0.016$; Fig. 4] and this decrease was similar for hits and misses [$−0.189 < $ all $\rho s < 0.171$, all $ps > 0.484$; Fig. 4]. Moreover, even though memory performance decreased over time across subjects, this effect was not significant [$\rho = 0.398, p = 0.127$].

We repeated the same analysis for each source to inspect whether the time-on-task effect was driven by some domain-general processes (see Supplementary Information for the results, "The time-on-task effect for each source" and Supplementary Fig. 14). We found that the level of sustained and selective attention significantly decreased as the time-on-task for the corresponding tasks increased but no change was found for the perception task. It is important to note that the perception task took about 8 minutes to complete, while the sustained and selective attention tasks took about 15 and 20 minutes respectively. Whether there would be a negative time-on-task if the perception task had been as long as the encoding and the other source tasks is unclear. Overall, as mentioned in the introduction, the time-on-task effect has been previously captured in various cognitive domains[24–26] and one could speculate that it is tied to a domain-general process (such as potentially mental fatigue) that is potentially shared between all the current sources. This is not in contrary to the logic of transfer learning as we, using the stepwise classification results, showed that there are some domain-general processes that are shared across all sources.

### The engagement of perception, sustained attention, and selective attention during encoding depended on the encoding success history

Next, we tested whether an event is more likely to involve higher levels of the underlying cognitive processes when it is preceded by a history of higher levels of engagement from those processes. As such, we found that the neural evidence of high levels of perception, sustained attention, and selective attention were higher for events preceded by a hit compared to events preceded by a miss [$F(1, 760) = 16.92$, $p < 0.001$, $\eta_p^2 = 0.021$; Fig. 5A].

While the previous encoding event's encoding success could influence the current encoding event's underlying processes, by the same logic, it can be inferred that the brain state associated with the current event may also carry over into the next. Specifically, trials with high levels of perception, sustained, and selective attention may be more likely to precede high than low levels of these sources on subsequent trials. Consistent with this hypothesis, we found that events preceding a hit were more likely to engage higher levels of the underlying cognitive processes relative to the events preceding a miss [$F(1, 760) = 6.01, p = 0.015, \eta_p^2 = 0.008$; Fig. 5B].

## Classification performance based on the number of sources included

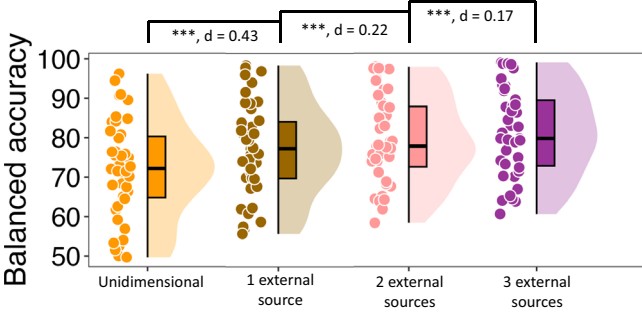

**Fig. 3 | Comparing the classification performance depending on the number of included sources.** Here, we have shown the item memory classification performance as a function of how many of the sources are included (averaged across all six possible orders) during classification. Specifically, adding each source significantly improved the classification accuracy [all $ts > 5.968$, all $ps < 0.001$, one – tailed, all $ds > 0.16$]. Circles reflect the data points of individual participants ($N = 43$). In the box plots, the minima represent the lowest data point within a condition, maxima represent the highest data point, centre represent the median value within the box, bounds of the box are the 25th and 75th percentiles, whiskers extend from the box to the minimum and maximum values that are not considered outliers, and percentile refers to the position of a data point within the distribution, with the box representing the middle 50% of data points between the 25th and 75th percentiles. The asterisks reflect statistically significant differences using one-tailed tests across conditions using Holm-Bonferroni corrections for multiple comparisons (\*\*\*$p < 0.001$) and the associated Cohen's d is shown for each comparison. Source data are provided as a Source Data file.

We repeated the same analysis for each source to inspect whether the history effect reflects some domain-general processes (see Supplementary Information for the results, "The history effect for each of the three sources" and Supplementary Fig. 15). We found that only during the sustained attention task, the associated brain state during an event was related to the success/failure of the previous/subsequent event. This indicates that the brain states for only certain tasks are likely to last beyond a single event and this phenomenon is not related to a domain-general process. One potential explanation is that the history effect observed in episodic encoding is driven by sustained attention processes. Given that sustained attention is necessary for memory formation, the encoding-related brain states are likely to last beyond a single event.

## Discussion

In this study, we investigated the contribution of brain states associated with several cognitive functions including visual perception, sustained attention, and selective attention to encoding-related brain states to better predict whether an event will be successfully encoded. The findings of this study provide invaluable insight into the multidimensional attributes of episodic memory that contribute to successful vs. unsuccessful encoding.

We found that investigating memory as a multidimensional process significantly increased our ability to predict memory outcomes as opposed to the more traditional approach of investigating memory as a unidimensional process. To understand why transfer learning is improving the results, it is critical to mention that each selected source task engages cognitive "subprocesses" that support episodic encoding. For example, some aspects of color perception and categorization (e.g., red vs. blue) likely underly performance in both the perception task and the episodic memory task, especially given the relevance of color in the to-be-encoded events. In this respect, transfer learning capitalizes on these shared characteristics[11–16] between the sources and the encoding-related activity to improve prediction. Importantly, the

ability to transfer the features learned in a source task to the memory task relies on the implicit assumption that there is an extent of context-invariance in the measured cognitive processes. For example, to successfully transfer the selective attention source to the memory data, it is essential that the neural correlates of selective attention processes are similar in the selective attention and memory tasks. However, inevitably, there will be an extent of unaccounted context-variance between each source task and the target task which could underestimate the extent the associated cognitive processes are engaged during memory encoding.

A conceptual question about transfer learning is why using the information from external tasks improves the prediction performance beyond what is achievable from within-task training and testing. To elaborate, to successfully transfer a source to the memory domain, the informative predictors from the source task are, themselves, present in the memory task data, as discussed above. One might argue that, by considering all the extracted features from the memory-task data, it should be theoretically possible for a memory-task trained classifier to learn from these informative features and have as high of a prediction performance as the multidimensional classifier. However, it is critical to note that the memory-task data on its own cannot provide how high or low each of these underlying processes are within each encoding event. To elaborate with an example, a hit trial could be "mixed" (i.e., some of the levels of perception, sustained attention, and selective attention could be low at some periods) and no matter how many features we extracted from the data, we would never be able to know that using the unidimensional approach. Thus, making events multidimensional using external sources allows us to predict how high or low each of the underlying processes are during each encoding event, helping us see the full picture and better predict encoding success.

The stepwise classification results raised important points that need to be further discussed. First, the reason memory classification improved at every step is that the selected source tasks are not redundant. This is not unexpected given that the source information is derived from activity associated with different cognitive tasks with different types of stimuli and task demands. Moreover, to empirically verify that each source task engaged discriminable patterns of neural activity, we performed a 3-class cognitive function classifier (i.e., perception, sustained attention, or selective attention, see Supplementary Information for more detail). The 3-class cognitive function classifier performed with 94.4%, and range of 69.6% to 98.4%, accuracy on average across participants, with empirical chance being 33.9%, after conducting permutation tests. Collectively, we can infer that each source task engages cognitive subprocesses that overlap with the encoding subprocesses and explains unique variance in episodic memory-related brain activity. Moreover, the diminishing return for adding the subsequent sources suggests that these cognitive functions are not completely independent from one another, with overlap in the patterns of neural activity underlying high vs. low performance across them. This is consistent with previous fMRI studies showing that several brain regions are "domain-general" structures engaged during performance of multiple types of tasks[6,7,33,34]. Similarly, previous EEG studies show that the oscillations of certain frequency bands reflect domain-general cognitive processes that are involved in several types of tasks[35,36]. Thus, when the second or the third source were included in the classification analysis, some of the characteristics they shared with episodic memory encoding were already captured when the first source was added, explaining the diminishing returns. Lastly, even though there were remarkable subject-to-subject variation in the results (as shown in Supplementary Fig. 10A, 10B), on average across all subjects, the order in which each source was added made no significant difference to the results. This suggests that each cognitive domain was similarly important in predicting memory success across but not within participants.

**Fig. 4 | The time-on-task effect for the engagement of perception, sustained attention, and selective attention processes during encoding.** The encoding events consisted of 240 stimuli presented during 4 blocks and 16 mini-blocks. The neural evidence of high levels of each source for hits and misses within each mini-block was averaged for each participant. The average of these evidence scores for each source across all participants is shown separately for hits and misses. The associated lines of best fit are shown as well. Source data are provided as a Source Data file.

We found that the extent to which these underlying cognitive processes were engaged during episodic memory encoding depended on the number of events participants had already encoded (i.e., time-on-task) as well as whether the prior event was successfully encoded (i.e., the encoding history). Regarding the time-on-task effect, we found the neural evidence for the underlying cognitive processes significantly decreased and while the memory performance also decreased over time, this decrease was not significant. Notably, previous neuroimaging studies of perceptual learning, sequential learning, and object naming have found that the activity in the corresponding brain regions decreases over time without compromising task performance, indicating improved efficiency[24–26]. However, other studies have shown that long periods of demanding cognitive activity could induce mental fatigue and negatively impact sustained attention[37,38]. While it is difficult to measure the extent to which neural fatigue and efficiency are separately impacting memory performance over time, it is plausible that both mechanisms are potentially contributing to the observed neural and behavioral effects.

The history effect found in this study is consistent with the idea that the brain states last beyond a single event and that the underlying cognitive functions engaged during encoding of an event hold a history of the associated brain states of the previous encoding event[27–30]. Our results are seemingly in contradiction to findings from Lohnas and colleagues who used intracranial EEG to show that non-recalled events with "good encoding history" (i.e., at least one of the two previous words was recalled) had worse encoding states based on the hippocampus activity than non-recalled events with "poor encoding history" (i.e., none of the two previous words were recalled)[21]. These results are consistent with the neural fatigue hypothesis[21], which postulates that it is more likely for encoding-related neural activity to decline than to rise after a sustained period of good encoding. However, unrecalled items were more common in this prior study than were misses in our study and as such, the majority of the subsequent misses in our study were preceded by "good encoding history" (based on their definition), preventing us to exactly replicate their analyses. And while the hippocampus activity was consistent with the neural fatigue hypothesis, the dorsolateral prefrontal cortex (DLPFC) activity showed opposite effects, consistent with the persistent encoding states. These differential effects on different brain regions makes it unclear how the scalp EEG activity would depend on the previous event's encoding success. Overall, since we focused on the single prior event's encoding success, our results are more in line with the idea that mnemonic brain states last beyond a single event. Having said that, the time-on-task effect

found on this study is consistent with the idea behind their study, suggesting the encoding performance will decrease after a sustained period of good encoding. Lastly, on a separate note, given that the encoding decisions are likely/unlikely context decisions and during the retrieval phase, an item memory decision was followed by two context memory decisions before the next event's item memory decision, response priming cannot account for this result.

Crucially, the idea of teasing apart a cognitive process into its cognitive components is not restricted to the episodic memory domain and the approach of this study also offers avenues for neuroscientists that are interested in other cognitive domains. In fact, one could potentially break down any cognitive function into its underlying cognitive processes to better understand that cognitive function. As an example, to better understand the underlying neural activity during threat detection (i.e., the target), using a similar approach to this study, one could investigate the involvement of processes related to visual perception, arousal, emotion, autobiographical memory, visual imagery, etc. (i.e., the sources). To support this idea, using our current dataset, we used selective attention as the target domain and investigated the contribution of visual perception, sustained attention, and episodic encoding processes during selective attention-related activity (Supplementary Fig. 13; see Supplementary Information for the results). Importantly, episodic memory did not explain unique variance above and beyond that explained by sustained attention and perception, suggesting that selective attention underlies episodic encoding more than episodic encoding underlies selective attention. Moreover, even though for episodic encoding, the selected sources contributed equally to encoding-related activity, this finding suggests that different sources could contribute differently to the target depending on the selected sources and the target.

The findings of this study should be interpreted in the context of a few limitations. First, it is unlikely that the three source tasks (or any other task) are completely "process-pure" and there are likely multiple cognitive processes that differentiate high and low trials in each of the tasks. For instance, earlier, we mentioned color perception and categorization as some of those potential subprocesses but there are likely additional subprocesses involved during some/all of the used tasks. Thus, pinpointing what exact subprocess for a given source task is producing the positive transfer effect (for each encoding event and for each participant) is beyond the scope of this study. In addition, other cognitive functions besides those investigated here, including executive function, working memory, emotion, etc., likely contribute to episodic encoding. Future studies could design other

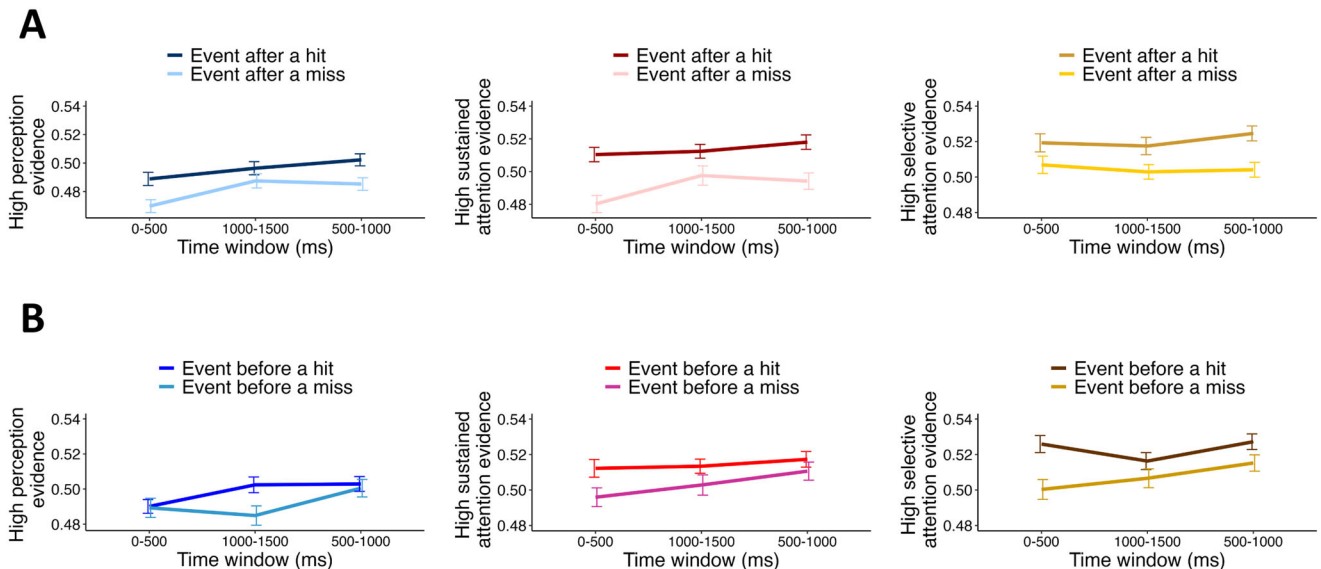

**Fig. 5 | Impact of encoding history on an event's underlying cognitive processes.** The level of engagement of the underlying cognitive processes during different encoding periods for events: (**A**) following a hit vs. following a miss and (**B**) preceding a hit vs. preceding a miss. The events are collapsed and averaged across hits and misses. The average level of engagement of each underlying cognitive process across all participants during early, middle, and late encoding periods is shown. We conducted Memory condition × Source × Time ANOVA for the associated analyses (see Methods for more details). The main effect of Memory Condition was significant for the effect of the prior event

$[F(1, 760) = 16.92, p < 0.001, \eta_p^2 = 0.021]$ and the effect of the subsequent event $[F(1, 760) = 6.01, p = 0.015, \eta_p^2 = 0.008]$ on the current event. It is worth mentioning that in both sets of ANOVA analyses, there was no Time × Source × Memory Condition interaction effect [all $Fs < 1.75$, all $ps > 0.176$, all $\eta_p^2 s < 0.005$] and the interpretation regarding the main effect of Memory Condition was one-sided. The associated error bars for each condition reflect standard error of the mean obtained across all participants ($N = 43$). Source data are provided as a Source Data file.

source tasks to investigate their involvement during episodic encoding. Although our control analyses support our assertion that the source processes (perception, selective, and sustained attention) were relevant to episodic encoding, in a future study, another good control would be to assess the contribution of neural activity from a task-irrelevant "sham" task (e.g. auditory perception). Furthermore, using functional magnetic resonance imaging (fMRI) could better elucidate the brain networks and areas that are involved when assessing the engagement of different cognitive functions during encoding. However, assessing the spatial information regarding the selected features was still informative (see Supplementary information "Importance of different brain areas in the classification results"). The last limitation that is worth mentioning is related to all studies that attempt to predict encoding success, and this study is no exception. Specifically, it is not definitive that a subsequently forgotten event was not actually encoded. It could be the case that an event was weakly encoded and unsuccessfully consolidated. In addition, participants might make response errors during retrieval (i.e., responding "new" when they intended "old"), This means some "misses" might actually be associated with successful encoding, which could reduce the classification accuracy. Altogether, it is important to acknowledge these limitations and note that for any memory prediction study, including this one, these factors could have a remarkable negative impact on the classifier's performance.

Critically, not only does this study shed light on the multidimensionality of memory, which is important for basic science, but it also opens avenues for future implications in terms of real-world interventions to improve memory. This multidimensional evaluation approach substantially improved prediction accuracy but more importantly, perhaps, it offers potential to personalize future feedback systems that could be implemented for real-world intervention applications. To elaborate, this multidimensional perspective could allow us to understand why a specific event is potentially not successfully encoded and provide the intervention accordingly. For instance, if the

intervention system detects memory failure mainly because of low sustained attention, suitable feedback potentially could be to take a break. Alternatively, if unsuccessful encoding is because of low selective attention, effective feedback could be to introduce salient cues to enhance attention. The findings related to the time-on-task and history effect could be important for memory intervention systems as well. To elaborate, as an individual encodes more events, and depending on the previous event's memory success, a multidimensional intervention system could adapt and adjust its prediction criterion to consider the encoding history and person's potentially increased efficiency. Thus, the ability to better predict memory success and provide personalized interventions and adapt the memory predictor system based on the time-on-task and history advances the field of cognitive neuroscience closer to the goal of designing an effective, real-time, memory-improvement system.

## Methods

### Participants

The participants consisted of 47, right-handed adults (22 men, 24 women, 1 non-binary) from ages 18 to 35. We excluded 4 participants: 3 of them were outliers (more than 3 standard deviations below the mean) in terms of performance in at least 2 of the tasks and the other person had post-traumatic stress disorder (PTSD) even though we had not realized that during initial screening. All participants were fluent in English and had normal or corrected vision. Subjects were compensated with $20/h and were recruited from the University of Texas at Austin and surrounding community. Except for the participant diagnosed with PTSD, none of the other participants reported any psychiatric or neurological disorders, vascular disease, or use of any type of medication that affects the central nervous system. All participants signed consent forms, and the entire study was approved by the University of Texas at Austin Institutional Review Board. Moreover, all participants completed 4 questionnaires including The Attention-Related Cognitive Errors Scale (ARCES)[39], Sleep Quality Assessment

(PSQI)[40], Center for Epidemiologic Studies Depression Scale Revised (CESD-R-20)[41], and Epworth Sleepiness Scale[42].

## Experimental tasks

It is important to mention that all the tasks were completed in a single session in one day. The episodic memory task (both encoding and retrieval stages) was always performed first followed by the three source tasks. The order of the three source tasks was counterbalanced across participants. It took roughly 45 minutes to perform the memory task, 8 minutes to perform the perception task, 20 minutes to perform the sustained attention task, and 15 minutes to perform the selective attention task.

## Visual perception task

In this experiment, we examined the level of perceptual processing using a task used often in the literature that also minimizes the influence of other cognitive processes[43–45]. Supplementary Fig. 1 shows the procedure. The task included 4 blocks where each block had 72 trials. For each trial, a black fixation cross was presented in the middle of the screen for 500 ms before the stimulus was shown for 500 ms. Each stimulus was a 7 × 7 grid of blue and red circles. For each grid, there were 27 circles with the dominant color and 22 circles with the other color. Half of the grids had more blue circles and the other half of the grids had more red circles. The order of shown grids was random for each participant. The subject had to decide whether there are more red or more blue circles in each grid by pressing 1 or 2 on the keyboard. Each stimulus was presented for 500 ms and the subject had 2 seconds to make a decision. There were 10 practice trials at the beginning and participants were instructed to keep their eyes in the middle of the screen for the whole duration of each trial. Within each individual, correct responses corresponded to the higher level of perception while incorrect responses corresponded to the lower level of perception. To perform binary classification and separate high vs. low levels of perception, we classified correct and incorrect responses.

## Sustained attention task

In this experiment, we examined the sustained attention level while minimizing the influence of activities related to other cognitive domains using Conjunctive Continuous Performance Test-Visual (CCPT-V)[46]. Supplementary Fig. 2 shows the procedure. The task included 4 blocks where each block had 200 trials. For each trial, the stimulus was a colored geometric shape shown at the center of the screen. The task included 16 different stimuli generated from all combinations of four colors (green, yellow, red, or blue) and four shapes (triangle, circle, square, or star). The target (red square) appeared on 30% of the trials (i.e., 240 trials). A red non-square appeared on 17.5% of the trials, a non-red square appeared on 17.5% of the trials, and for the remaining 35% of the trials a shape that was neither red nor squared appeared. Each stimulus was presented for 100 ms and was separated from the next stimulus by an interstimulus intervals (ISI) of 1000, 1250, or 1500 ms. Each ISI appeared in 1/3rd of the trials. The stimuli and ISIs were selected randomly. There were 10 practice trials at the beginning and participants were instructed to press the space bar to respond, as soon as the target appeared and to withhold from responding to all other stimuli. Within each individual, faster response times for correct responses correspond to the higher level of sustained attention. To perform binary classification and separate high vs. low levels of sustained attention, we only used correct target trials with the fastest 40% corresponding to "high" and the slowest 40% to "low" sustained attention. The top 40% and bottom 40% cutoff ensured that we are including as many trials as possible while having a clear boundary between "high" and "low" trials. We did not classify correct vs. incorrect decisions since the performance was on the ceiling level.

## Selective attention task

In this experiment, we examined the selective attention level while minimizing the influence of other cognitive-related activities. Supplementary Fig. 3 shows the procedure. This task is known as a Spatial Cued-Identification Task (SCIT) which is a spatial cued task with exogenous cues[46]. To elaborate, the fixation display included a black cross in the middle of the screen and two black rectangles to the right and left of fixation. The cueing display was similar to the fixation display but one of the rectangles would brighten briefly. The target display consisted of a black circle placed on the fixation display and centered in one of the two rectangles. On each trial, the fixation display appeared for 1000 ms, then the cueing display were shown for 200 ms. After a 100 ms ISI, the target was presented for 100 ms. 75% of trials were valid (i.e., the target appeared where the rectangle had brightened earlier) and 25% were invalid (i.e., the target appeared where the other rectangle was), randomly intermixed within a block. The task consisted of 4 blocks of 100 trials each. There were 10 practice trials at the beginning and participants were instructed to select the rectangle that contained the target as soon as possible while keeping their eyes on the fixation cross for the whole duration of each trial. Within each individual, faster response times for correct responses correspond to the higher level of selective attention. To perform binary classification and separate high vs. low levels of selective attention, we only included the trials that were correctly identified in the valid condition with the fastest 40% corresponding to high and the slowest 40% to low selective attention. The top 40% and bottom 40% cutoff ensured that we are including as many trials as possible while having a clear boundary between "high" and "low" trials. We did not classify correct vs. incorrect decisions since the performance was on the ceiling level. This is because the valid and invalid conditions require different cognitive operations and including the trials from both conditions would negatively impact the classification procedure.

## Episodic memory task

The episodic memory (i.e., target) task was designed to be complex, much like those in real life, and include multiple cognitive facets including visual perception, sustained attention, and selective attention[4,47,48]. 360 images of objects were chosen from the Bank of Standardized Stimuli (BOSS) datasets[49,50] and we turned them into grayscale. Each grayscale object was presented in the middle of the screen with gray background. Scenes and color squares were shown to either side of the object. The locations of the context features (i.e., color or scene) were counter-balanced across blocks so that they were presented an equal number of times on either side across subjects. The scenes consisted of color photos of a studio apartment, cityscape, or island. The colored squares were red, green, or brown. Each context and object pictures spanned a maximum horizontal and vertical visual angle of around 3°. 240 of the objects were shown during encoding while during retrieval, 120 new objects were presented in addition to the 240 previously presented objects. Study and test objects were counterbalanced across participants.

Supplementary Fig. 4 shows the procedure used at the study and test stages. Before each stage, participants were provided instructions and given 10 trials to practice. For each encoding trial, participants were instructed to attended to either the colored square or the scene, which served as the attended context for that trial. Participants were asked to make a subjective yes/no judgement about the relationship between the object and one of the colored square (i.e., is the color likely for this object?) or the scene (i.e., is this object likely to appear in the scene?). The study phase consisted of four blocks where each block included four mini-blocks, each of which consisted of 15 trials. Before beginning of each mini-block, a prompt was shown (e.g., "Now you will assess whether the color is likely for the object" or "Now you will assess whether the scene is likely for the object"). Furthermore, each trial had

a reminder prompt presented below the pictures during study trials (see Supplementary Fig. 4).

In the test stage, both old and new objects were shown. Similar to the study phase, each object was presented with a colored square as well as a scene. For each object, the participant initially had to decide whether it was old or new. If the participant decided the object was new, the next trial began after 2000 ms. If they decided it was old, they had to make two additional assessments about whether each of the color and the scene matched or did not match the ones initially presented with the same object during the study phase. The order of the second and third questions was counterbalanced across participants. For old objects, the pairing was set in a way that an equal number of old objects were shown with: (1) both color and scene matching those presented at the study stage, (2) only the color matching, (3) only the scene matching, and (4) neither color nor scene matching. In total, there were four study and four test blocks. To perform binary classification, we separated subsequently remembered items (i.e., "item hits") from subsequently forgotten items (i.e., "item misses"). Furthermore, to perform binary classification and separate high vs. low levels of attended context memory encoding, we separated correct context decisions (i.e., the context was correctly identified as a match or a mismatch later at retrieval) from incorrect context decisions (i.e., the context was incorrectly identified as a match or a mismatch later at retrieval).

Notably, all the analyses were performed for classifying brain states associated with item memory (i.e., item hits vs. item misses) as well as attended context memory (i.e., correct vs. incorrect context decisions). Importantly, item and context encoding processes are correlated and we did not make any specific prediction about item and context encoding being different for any of our hypotheses in this study. Thus, we focused on the results obtained from item memory classification analyses. Regardless, the findings of the analyses related to attended context memory classification were very similar to the ones related to item memory classification (see Supplementary Information section "The classification results to predict attended context memory success" and Supplementary Fig. 12 for the results).

## EEG recording
Continuous scalp-recorded EEG data was collected from 31 electrodes using the Brain Vision ActiCAP system. Electrode position reflected the extended 10–20 system[51]. Electrode positions consisted of: Fp1, Fz, F3, F7, FT9, FC5, FC1, C3, T7, TP9, CP5, CP1, Pz, P3, P7, O1, Oz, O2, P4, P8, TP10, CP6, CP2, C4, T8, FT10, FC6, FC2, F4, F8, and Fp2. External right and left mastoid electrodes were used for offline referencing. Two additional electrodes recorded horizontal electrooculogram (HEOG) at the lateral canthi of the right and left eyes and two electrodes placed inferior and superior to the right eye recorded vertical electrooculogram (VEOG). The sampling rate of EEG was 500 Hz without high or low pass filtering.

## EEG preprocessing
Offline analysis of the EEG data was performed in MATLAB using the EEGLAB[52], ERPLAB[53], and FIELDTRIP[54] toolboxes. For the episodic memory data, each epoch was baseline corrected according to 600 to 400 ms before the stimulus onset while for the source tasks, each epoch was baseline corrected according to 400 to 200 ms before the stimulus onset. We used an earlier baseline for the episodic memory data to be able to potentially investigate the pre-stimulus periods for the encoding task, not presented here. The sampling rate was reduced from 500 Hz to 250 Hz and bandpass filtering was applied to the data to include 0.05 to 80 Hz frequencies. We first used visual inspection to removed trials that had muscle artifacts, electrode artifacts, and sweat artifacts. Subsequently, we ran an independent component analysis (ICA) on all head electrodes to identify ocular artifacts (i.e., blinks and horizontal eye movements). Components associated with eye

movements were removed by visually inspecting the topographic component maps as well as the component time course. If the data had a noisy electrode (e.g., more than 40% of the epochs had to be removed), it would be removed and interpolated using the surrounding electrodes to estimate the noisy electrode's activity. Next, we automatically removed the epochs that were showing extreme voltages (higher than 150 microvolt) for at least 2 electrodes and used visual inspection to remove the epochs that still contained any type of artifact. Lastly, we ran the time frequency procedure by using Morlet wavelets at 5 cycles with 78 linearly spaced frequencies from 3 to 80 Hz. Each epoch was then down sampled to 50 Hz[52].

## Transferring the information from a source to the target
Using the EEG recordings from the sources, we trained binary high vs. low classifiers specific to those cognitive functions. During each time window of interest (more detail in the last paragraph of this section), we used the voltage (with sampling rate of 250 Hz, or 4-ms for each time bin) and the power of different frequency bands including theta (6–8 Hz), alpha (8–12 Hz), beta (13–30 Hz), and gamma (35–80 Hz) using time-frequency representation (with sampling rate of 50 Hz, or 20-ms for each time bin). We used both voltage and power as extracting as much information as possible from the neural activity allows us to have a stronger prediction power (see the Supplementary information, "Importance of different frequency bands and voltage in the classification results" for more details). We log-transformed the power of each frequency to ensure the scale of the power values of different frequencies are at similar ranges. We extracted features based on common spatial patterns (CSP) as it has been commonly and effectively used in the transfer learning literature[17,18,20,31,32,55,56] (see below for description of CSP). For the time window of interest for classification, we extracted CSP-based features for the time series of voltage as well as the power of theta, alpha, beta, and gamma frequency bands, leading to 150 CSP-based features (more details in the next paragraphs). For feature selection and training the classifier, we used the recommendations mentioned in[4]. Specifically, it has been shown that feature selection using the Sequential Forward Selection approach (also known as the "wrapper" method) is optimal in terms of the associated classification performance. However, given that this approach performs classification (using numerous different sets of features) as part of the feature selection, it is quite time-consuming, and it is not practical to search through all 150 features to find the best set of features. Thus, we first "filter" a certain number of features and then use the wrapper method to find the best set of features. We evaluated how much each extracted feature (i.e., CSP-based feature extracted from the voltage or power data) can be useful for binary classification by calculating the associated Fisher score. As recommended in[4], the number of features to filter and the number of features to select among them using the wrapper method are the two parameters that should be chosen based on the researcher's preference for performance and running time. In this study, we found that filtering the top 40 features and selecting the best 5 features among them with the wrapper method ensured the optimal classification performance with reasonable running time (Supplementary Fig. 5). Moreover, given that the performances across different classifiers are fairly similar, and given that this study's analyses are computationally time-consuming, we used naïve Bayes to train the classifier due to its fast performance[4].

Importantly, we selected the top 5 features (among the 150 extracted features) to ensure the classification performance would be high without sacrificing the running time. However, with the current feature selection methodology, it is possible that the multidimensional approach increases the performance because potentially predictive features from the memory task were excluded during feature selection and thus the current unidimensional performance is an underestimate

of what is achievable. To rule out this possibility, for 10 randomly selected participants, we increased the number of selected features (spanning from 5 up to the full 150) to inspect how much the classification performance for both approaches would change (Supplementary Fig. 6). We found that the classification performance increased for both approaches as the number of selected features increased up until the classifier started to suffer from overfitting. Critically, even at the peak performance (i.e., around 50 selected features), the unidimensional classification did not perform as well as the multidimensional approach with only 5 selected features, suggesting the advantage of the multidimensional approach is not simply due to the current feature selection approach (more details in the discussion).

For all the classification analyses in this study, we used the synthetic minority oversampling technique or "SMOTE" to ensure there would be equal numbers of trials for both conditions and the classifier would not have a bias to label trials as the class with the majority of the trials[4]. Moreover, to reduce over-fitting, we used 5-fold nested cross-validation to train and test the classifier. In more detail, at each outer fold, 1/5th of the trials would be totally left out and they would never be used during training for that outer fold. The classifier's parameters (i.e., the top 5 features and transfer learning parameters; more detail in Supplementary Fig. 7A) would then be optimized by running 5-fold cross-validation on the remaining 4/5th of the trials. Once the classifier was optimized (i.e., the best set of features were obtained), we would test the classifier's performance on the 1/5th of the trials that were totally left out for the entire training process of that outer fold. We repeated the same process 5 times (i.e., leaving out 1/5th of the trials each time) so that every trial was left out once and we could measure the classifier's performance on the left-out trials without any inflammation. Moreover, we used balanced accuracy to evaluate the classification performance. Balanced accuracy is the average between sensitivity (i.e., true positive rate) and specificity (i.e., true negative rate) and it considers the imbalance between the number of trials in different conditions[4]. Since in this study, we used SMOTE to balance the number of trials for hits and misses, the balanced accuracy would theoretically be the same as typical accuracy (i.e., the ratio of correct predictions across both conditions).

To transfer a source domain to the target domain, one could train a classifier to separate the high and low levels of the source and test the same classifier on the target data. However, it deserves mention that the target and source usually represent data from different participants, different recording sessions, or like this study, different tasks. Thus, if for the target, one extracts the same effective features that separated high and low levels of the source and uses the same trained classifier, with no adjustment, to make a prediction for the target events, the classifier's output is usually not reliable. To illustrate with a simple hypothetical example, and this is not reflective of how the proposed algorithm exactly works, imagine trying to determine whether the level of perception is high during an encoding event. Using the perception data, we realize that higher posterior theta activity is associated with higher perception. In that case, if the posterior theta power was higher than $40 \frac{\mu v^2}{Hz}$, the trial is associated with high perception. Hypothetically speaking, it might be the case that the posterior theta power is always in the range of 10 to $30 \frac{\mu v^2}{Hz}$ during the encoding experiment. In that case, if we test the same classifier on the encoding data, it will predict that the perception was low during all events. As a result, it is important to make an adjustment to the decision boundary to make an accurate prediction about the perception level during encoding. For example, after using transfer learning and considering posterior theta activity during successful and unsuccessful encoding, we might realize that high perceptual activity during encoding is associated with posterior theta activity that exceeds $15 \frac{\mu v^2}{Hz}$ rather than $40 \frac{\mu v^2}{Hz}$. Thus, to effectively leverage the information from a source to the target, it is critical to use transfer learning algorithm (Supplementary Fig. 8).

There have been several approaches used in the literature for transfer learning, but one of the most common approaches is based on common spatial patterns (CSP)[17,18,20,31,32,55,56]. While there is no general agreement about the specific optimal CSP-based approach, we used the methodology used in[20] which is known as Regularized CSP or RCSP. We used this method because of its effective performance and simplicity to implement and interpret. Briefly, the typical CSP algorithm designs spatial filters that maximize the variance difference between the trials of the two distinct classes. The spatial filters are obtained by using the concept of eigen value decomposition after solving the following optimization problem:

$$w = \arg\max \frac{w^T \overline{C_{\{1\}}} w}{w^T \overline{C_{\{2\}}} w} \tag{1}$$

For example, for a 200-ms time window of interest for trial $X$, we'll have a time series of 50 voltage (in 4-ms time bins) and 10 power (in 20-ms time bins) values at different frequency bands. Thus, trial $X$ is represented by a 31-by-10 (for the theta, alpha, beta, and gamma frequency bands) or a 31-by-50 (for voltage) matrix. The associated 31-by-31 covariance matrix $C$ will be computed as:

$$C = \frac{XX^T}{trace\left(XX^T\right)} \tag{2}$$

$\overline{C_{\{1\}}}$ and $\overline{C_{\{2\}}}$ represent the average covariance matrices of the trials of each two class. Moreover, given that our EEG dataset has 31 electrodes,

$$w = \begin{bmatrix} w_{1-1} & \cdots & w_{1-31} \\ \vdots & \ddots & \vdots \\ w_{31-1} & \cdots & w_{31-31} \end{bmatrix}$$ represents a 31-by-31 matrix where the 31

rows of $w$ represent 31 spatial filters that are designed to optimally project the data into a new space in which the data at every projected electrode is a linear combination of the data across all original electrodes. Specifically, for the first row of $w$ (i.e., $w_1 = [w_{1-1}, w_{1-2}, \ldots, w_{1-31}]$), the variance of the time series (either voltage or power) of the associated projected electrode are maximally high for trials of class 1 and minimally low for trials of class 2 (see Supplementary Information "Importance of different brain areas in the classification results" and Supplementary Fig. 9). For the next rows of $w$, the variance of the projected electrodes keeps decreasing for trials of class 1 while increasing for trials of class 2. Thus, for the last row of $w$ (i.e., $w_{31} = [w_{31-1}, w_{31-2}, \ldots, w_{31-31}]$), the variance of the time series of the associated projected electrode are minimally low for trials of class 1 and maximally high for trials of class 2. In this study, we combined two regularization approaches of the CSP filters to perform the transfer learning. Specifically, separately for each source task, 31-by-31 matrices $\overline{C_{\{1\}}^s}$ and $\overline{C_{\{2\}}^s}$ are obtained from that individual source task while 31-by-31 matrices $\overline{C_{\{1\}}^t}$ and $\overline{C_{\{2\}}^t}$ are obtained from the target task. Next, the two following optimization problems should be solved:

$$w = \arg\max \frac{w^T \overline{C_{\{1\}}^{reg}} w}{w^T \overline{C_{\{2\}}^{reg}} w + \beta w^T D_1 w} \tag{3}$$

$$w = \arg\max \frac{w^T \overline{C_{\{1\}}^{reg}} w}{w^T \overline{C_{\{2\}}^{reg}} w + \beta w^T D_{31} w} \tag{4}$$

should be solved. In this equation:

$$\overline{C_{\{i\}}^{reg}} = \alpha.\overline{C_{\{i\}}^s} + (1-\alpha).\overline{C_{\{i\}}^t}, \ i = 1, 2 \tag{5}$$

$$D_1 = \begin{bmatrix} \frac{1}{w_{1-1}^2} & \cdots & 0 \\ \vdots & \ddots & \vdots \\ 0 & \cdots & \frac{1}{w_{1-31}^2} \end{bmatrix} \quad (6)$$

$$D_{31} = \begin{bmatrix} \frac{1}{w_{31-1}^2} & \cdots & 0 \\ \vdots & \ddots & \vdots \\ 0 & \cdots & \frac{1}{w_{31-31}^2} \end{bmatrix} \quad (7)$$

$D_1$ and $D_{31}$ are each a 31-by-31 diagonal matrix with the diagonal elements being the inverse squared values of the first and last spatial filters ($w_1$ and $w_{31}$) respectively. $w_1$ and $w_{31}$ are selected as they are the two spatial filters that best discriminate the variance between the trials of the two classes. In the optimization equation, $\alpha$ and $\beta$ are scalar regularization parameters that need to be chosen appropriately during cross-validation, with $\alpha$ denoting how much each of the source and target tasks will drive the regularized spatial filters and $\beta$ reducing the impact of inappropriate channels for the target domain. The transfer learning process was performed for the voltage as well as the power values of theta, alpha, beta, and gamma frequency bands. And separately for each 4 frequency bands and the voltage, the output (i.e., the variance from the projected time series) from the first 15 rows (where the variance of class 1 trials are higher than class 2 trials) and the last 15 rows (where the variance of class 1 trials are higher than class 2 trials) of the associated $w$ matrices were used, leading to $30 \times 5 = 150$ features. Note that $\alpha = 0$ and $\beta = 0$ would result in the typical CSP approach for the single-task classification of the target task which totally ignores the information from the source. Thus, this optimization problem tries to improve the performance by utilizing the gained information from the source domain to add to what could already be obtained by only focusing on the target task. It is critical to note that the same principles would apply to transfer the information from one participant to another. In that case, the source would be the information from the "source subject" and the target would be the information (from the same or even a different task) from the "target subject". Since we did not have any specific hypothesis regarding between-subject classification, we did not perform any cross-subject classification analysis.

To take advantage of EEG's high temporal resolution, we investigated how high perception, sustained, and selective attention are at different encoding periods during an event. To elaborate, we first found the optimal 200 ms time window that could optimally classify high and low states associated with each source across participants. We then transferred the information from each source to different 200 ms periods of each encoding event. In more details, we first transferred the information from each source to the [0 200 ms] period during each encoding event. We then transferred the information from each source to the [100 300 ms] period during each encoding event. We kept on transferring the information from each source to the different periods of each encoding event (with the last period being [1300 1500 ms] during an encoding event). Thus, during each encoding event, we assessed the level of perception, sustained attention, and selective attention during 14 different 200 ms time windows including [0 200 ms], [100 300 ms], [200 400 ms], …, [1300 1500 ms]. An illustration of this approach is shown in Supplementary Fig. 7B. It is worth mentioning that the transfer learning algorithm requires the source and target to have the same length of data (in terms of the number of time points) to be able to transfer the source information to the target data. This is because the same set of CSP-based features (extracted from a 200 ms period during a source task) needs to be extracted from the target data and this means for each transfer learning procedure, we selected a 200 ms period during encoding. Having said that, if one

wants to transfer a longer period of source to a shorter period of the target, they can use moving averaging of the data to reduce the number of time samples of the source and then perform the transfer learning analyses.

## Integrating the information from all sources

After using transfer learning to obtain the levels of perception, sustained attention, and selective attention at 14 different time intervals during encoding for each event, it is essential to integrate these pieces of information to determine their collective ability to predict memory success or failure. While there have been some EEG transfer learning studies using multiple source domains for fatigue assessment, emotion recognition, and motor imagery[15,17–20], this is a relatively new field especially in the neuroimaging field and cognitive neuroscience area. Since there are only a few EEG studies on this topic with them being published in 2020 or later, there is little consensus on the optimal approach for integration of the information from different sources. We used the approach from a prior EEG study due to its effective performance as well as its simplicity to implement[19]. In more detail, each source classifier (i.e., for determining the level of perception, sustained attention, or selective attention) produced a score at 14 different time windows during each encoding event showing how the level of that associated source was during that period of encoding. These scores include $\text{score}_{\text{per}[0-200]}$, $\text{score}_{\text{per}[100-300]}$, $\text{score}_{\text{per}[200-400]}$, …, $\text{score}_{\text{per}[1300-1500]}$, for perception level at different encoding periods, $\text{score}_{\text{sus}[0-200]}$, $\text{score}_{\text{sus}[100-300]}$, $\text{score}_{\text{sus}[200-400]}$, …, $\text{score}_{\text{sus}[1300-1500]}$, for sustained attention level at different encoding periods, and $\text{score}_{\text{sel}[0-200]}$, $\text{score}_{\text{sel}[100-300]}$, $\text{score}_{\text{sel}[200-400]}$, …, $\text{score}_{\text{sel}[1300-1500]}$, for selective attention level at different encoding periods.

To collectively determine memory success for each encoding event, these scores were integrated by performing a linear regression. In more detail, the final memory score, by considering all the scores for the sources at all the 14 encoding time intervals will be computed in the following way:

$$\text{Memory score} = \sum_{t=1}^{14} k_{1-t}\,\text{score}_{\text{per}(t)} + \sum_{t=1}^{14} k_{2-t}\,\text{score}_{\text{sus}(t)} + \sum_{t=1}^{14} k_{3-t}\,\text{score}_{\text{sel}(t)}$$

(8)

Where $k_{i-j}$ denotes the regression coefficient for source $i$ (among perception, sustained attention, and selective attention) and time interval $j$ (among [0–200 ms], [100–300 ms], [200–400 ms], …, [1200–1400 ms], [1300–1500 ms]). If the predicted memory score for an event exceeds 0.5, it will be labeled as a hit and otherwise, it will be labeled as a miss. This technique associates larger regression coefficients with sources and periods that are particularly important for determining memory success and smaller coefficients to the sources and periods that are not as important for determining memory success. As a result, the linear regression coefficients provide critical insight into which cognitive functions during which encoding periods are particularly essential for memory formation for a participant (see Supplementary Fig. 10A, 10B).

## Inspecting the time-on-task effect on the levels of visual perception, sustained attention, and selective attention

In these analyses, we looked at the position in which an event was presented among the 240 presented encoding trials. For each encoding event, we took an average of the level of perception, sustained attention, and selective attention throughout the 14 time windows during the encoding period (i.e., across the entire 1500 ms after the event's onset) to end up with an average score for each source. Thus, we ended up with 240 (i.e., the number of encoding events) evidence scores for perception, 240 evidence scores for sustained attention, and 240 evidence scores for selective attention and we investigated how much each evidence scores change from the 1st encoding event

until the 240th encoding event for hits and misses separately. To inspect how the evidence scores for the three sources change from the 1st event to the 240th event for hits and misses separately, it would be ideal to have the evidence scores for every single trial number for both hits and misses for every participant. But this is simply not possible and is a limitation of this analysis. For example, for a participant, trials [4, 23, 76, …, 203, 239] were removed during preprocessing, trials [1, 2, 6, 8, 9, …, 237, 240] were hits, and trials [3, 5, 7, 11, …, 231, 238] were misses. And these indices will be different for each participant. For this analysis, however, it is necessary that each condition has source scores for all 240 trial numbers for all participants. In this regard, for any trial number (for hits and misses separately) that did not have evidence scores, we used the evidence scores associated with the closest trial number before that event. To elaborate, for a specific participant, suppose two misses are happening at trial numbers $i$ and $j$ and none of the trials in between (i.e., trials $i + 1, i + 2, …, j − 2, j − 1$) are misses. To all of those trials in between, we would associate the evidence scores of trial $i$ for that participant (Supplementary Fig. 11). We also tried associating those trials the evidence scores of trial $j$ and also the average of the evidence scores of trials $i$ and $j$ and the results were very similar. Importantly, the 240 trials were shown in 4 separate blocks where each block consisted of 4 mini-blocks. For making the visualization and analyses easier, we divided these 240 trials to 16 mini-blocks and showed the average evidence for each source within each of these 16 mini-blocks (Fig. 4). To measure how much the neural evidence for a source for each memory condition changes throughout the encoding experiment, we computed the slope of evidence values against the number of trials encoded across subjects. This was done by calculating the Pearson correlation between the mini-block number and the associated average evidence score for that source across participants. In addition, to determine whether the time-on-task affects hits and misses separately (i.e., hits and misses show different slopes), we calculated the Pearson correlation between the mini-block number and the difference of associated average evidence scores between hits and misses for that source across participants. In other words, we tested whether the evidence scores for each source for hits and misses have significantly different slopes. Lastly, to determine whether the number of misses increases throughout the encoding experiment, we calculated the number of misses during each mini-block across all participants and then computed the slope of number of misses per mini-block against the number of mini-blocks across subjects. This was done by calculating the Pearson correlation between the mini-block number and the associated average number of misses across participants.

### Investigating the effect of encoding success history

We first investigated whether the encoding success history of the previous event (i.e., whether the previous event was a hit or a miss) could impact the underlying processes occurring during the current event (collapsed and averaged across hits and misses). Given that the brain states are not transient and potentially last beyond a single event, we expect this endurance to last more than a few hundred milliseconds. Moreover, there is no specific prediction that a particular period during encoding such as [300–400 ms] will be affected the most. Thus, it is reasonable to investigate the encoding period in 3 chunks of 500 ms (i.e., early encoding [0–500 ms], middle encoding [500–1000 ms], and late encoding [1000–500 ms]) instead of shorter periods. For each participant, we first averaged the evidence scores for each three sources for hits after a hit and averaged the evidence scores for each three sources for misses after a hit and then took an average to end up with average evidence scores of the three sources in three encoding periods for events after a hit. This was done similarly for events after a miss. In other words, each participant had one average evidence score for each source (i.e., perception, sustained attention, selective attention) during each encoding period (i.e., early, middle,

late encoding) separately for events after a hit and events after a miss. We submitted these neural evidence scores for the three sources at different encoding periods to a memory condition (events after a hit vs. events after a miss) × source (perception, sustained attention, selective attention) × time (early encoding [0–500 ms], middle encoding [500–1000 ms], late encoding [1000–1500 ms]) ANOVA. We performed a similar analysis to compare events preceding hits vs. misses.

### Reporting summary

Further information on research design is available in the Nature Portfolio Reporting Summary linked to this article.

## Data availability

All data and materials used in the analysis are available on https://osf.io/8vfy4/. Source data are provided with this paper.

## Code availability

The codes used for the analysis are available on https://osf.io/8vfy4/.

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

## Acknowledgements

We sincerely thank J. A. Lewis-Peacock, F. Pestilli, and T.-P. Jung for helpful discussion. We also highly appreciate the help of C. Nyan and S. Ram for assisting with the EEG data collection. This work was supported by National Science Foundation grant 1125683 to A.D., National Science Foundation grant BCS-1850802 to A.D., Ruth L. Kirschstein NRSA Institutional Research Training (from the National Institutes of Health) grant 5T32AG000175 to A.D., and Sigma XI Aid of Research grant G20221001-4163 to S.M.

## Author contributions

Conceptualization: S.M., A.D. Methodology: S.M., A.D. Investigation: S.M., A.D. Visualization: S.M. Funding acquisition: S.M, A.D. Project administration: A.D. Supervision: A.D. Writing—original draft: S.M. Writing—review and editing: S.M., A.D.

## Competing interests

The authors declare no competing interests.
