## [Transparent Peer Review file · Nature Communications]

Using machine learning to simultaneously quantify multiple cognitive components of episodic memory

Corresponding Author: Dr Soroush Mirjalili

Version 0:

Reviewer comments:

Reviewer #1

(Remarks to the Author)

In the current study, the authors use a “transfer learning” approach and report that by leveraging neural information (as measured by scalp-EEG) in multiple tasks that are thought to measure separate cognitive processes, subsequent memory performance on a memory task can be more accurately predicted relative to relying solely on neural information in the memory task. The paper frames well-motivated research questions and the study draws on innovative methods. At the same time, several key issues need to be addressed and doing so has the potential to strengthen support for the central conclusions and thus the ultimate impact of the work.

Major Comments:

1. Feature-selection procedure and its effects on the results

a. The reported analyses demonstrate the impact of having additional predictors from other tasks. However, given the approach to feature selection, one might worry that this transfer occurs because potentially predictive features from the memory task were excluded during feature selection and thus the memory-to-memory prediction is an underestimate of what is achievable. That is, could the finding that direct classification in the memory task is lower than classification when also including across-task predictors reflect the arbitrarily (low) number of features used to predict performance? While the analytic pipeline begins with 150 features, they are then reduced to the top 5 features. Some of the 145 excluded features might be/likely are informative, and their inclusion might boost unidimensional performance to be close to that of the multidimensional approach. To address this question, it would be helpful to report a plot of classification/prediction performance a function of the number of features included in the model, spanning from 5 up to the full 150 (using L2 or some other form of regularization to handle collinearity). Such a plot should be reported for both the unidimensional (memory-to-memory) and multidimensional approaches.

b. The results in the Supplementary Figure 9 do not speak against the above speculation. Assuming that those results were generated with the same feature-selection procedure, the fact that the top 5 features selected from the memory task did not add additional explanatory power to the selective attention classification while the reverse (attention-to-memory transfer) did increase classification suggests that the Top 5 features in the memory task do not overlap meaningfully with features that explain performance in the attention task. However, it remains a possibility that the Top 5 features in the other tasks are lower ranked features in the memory task and thus they were excluded from the latter; the “transfer learning” approach effectively adds them back in as additional predictors and performance increases. From this perspective, this improvement needn’t depend on obtaining these features from other tasks; rather, the features may well exist in the memory task itself and if these additional features from the memory task are included in the model (by including more than 5 features), then a similar performance increase might be obtained.

c. Since feature-selection was performed for each participant, can the authors comment on the extent of feature overlap across participants? If there is marked divergence across participants, given that some features (e.g., oscillation bands) have been tied to a variety of theoretical constructs, then it would be helpful to comment on what does it mean for different participants to have different sets of top features.

d. A related conceptual question worth discussing is why “transfer learning” might be expected to improve performance over within-task training and testing. Specifically, in the “transfer learning” approach the ultimate evaluation is to test the out-of-task classifiers on predicting memory performance. For this transfer to occur, it must mean that the informative predictors from the other tasks are, themselves, present in the memory task data. Thus, shouldn’t it be theoretically possible for a memory-task trained classifier (with more than 5 features) to learn from these informative features and thus lead to further performance boosts? What is it about out-of-task model training that leads to learning over these features that would not be

possible from within-task model training?

2. As presented in the manuscript, both the rationale and the analyses conducted appear to rest on an implicit assumption that the three source tasks are process-pure, which is unlikely to be true and needs further comment. That is, there are likely multiple cognitive/neural processes that differentiate high and low trials in each of the tasks, including those framed by the authors but also additional processes. How do we know which of these processes for a given source task is producing the positive transfer effect? That is, what does the positive transfer tell us about the particular cognitive processes that differentiate successfully vs unsuccessfully encoded trials?

3. Related to the previous point, the source tasks are minimally analyzed. It is unclear which components of each source task contribute to the classifier performance (e.g., is there a time-on-task or trial-history effect in some/all of the source tasks as well? For trial history effects, one could imagine that high/low performance trials could partly reflect differences in response priming/conflict, amongst other likely processing demands present in each task). If time-on-task and/or trial history effects are caused by some domain-general mechanisms independent of the specific cognitive processes you aim to measure, wouldn't they also affect the source tasks? How might this impact the logic of the transfer learning aspect of the work?

4. The ability to transfer the features learned in the source tasks to the memory task relies on the assumption that there is some degree of context-invariance in the measured cognitive processes. As a specific example, the neural correlates of selective attention would need to be similar in the selective-attention task and the memory task to observe "transfer learning". Could unaccounted context-variance lead to an underestimate of the extent to which specific cognitive processes are involved in memory encoding? This point may be worth considering in the Discussion.

5. Some details about the transfer-learning methods need further clarification:

- a. Please clarify why $D1$ is based only on the first row of w , which seems to be a column vector, so the first "row" of it is just one value. Does that mean the diagonal elements in $D1$ are all the same because other rows of w are not considered? The notation of " w_1 " is also confusing because subscript has been used to denote class rather than row number up to this point.
- b. The description of C_i is also not clear. They "represent the average covariance matrices of the trials of each class where the trials are matrices of EEG voltages across time bins and electrodes"; what does "time bins" mean in this context? Were only raw voltages considered but not spectral features?
- c. Please specify the dimensionality of all variables in the equations, along with what those dimensions represent (e.g. is w a #channel-by-1 column vector?). Perhaps consider choosing different indexing methods when referring to different dimensions.

6. When framing the work in the Introduction, at times, some claims about the state of the literature appear overly strong or narrow. Specifically:

- a. On line 67, the claim that "no study has investigated the involvement of multiple cognitive functions during episodic encoding simultaneously" is a bit too strong. For example, multiple prior studies have examined the effects of various forms of attention on episodic encoding.
- b. Embedding the trial-history effect in the temporal-context framework seems unnecessarily narrow, as other mechanisms can produce similar effects (e.g. stimulus/response priming).
- c. On line 77, the term "orthogonal tasks" would benefit from increased definition. Is the fact that the features transfer between tasks an apparent contradiction with the term "orthogonal"?

7. For trial-history effects (Fig 5), how were performance differences on the current trial controlled for? In other words, can one be sure that the effect is driven by performance on neighboring trials over and above the presumably different percentages of hits and misses in the current trial?

Minor Comments:

1. Perhaps this reflects a confusion, but did the EEG baseline (-400 to -600 ms) for episodic memory exceed the ITI for some trials (-350 to -700 ms)? The same (-200 to -400 ms) appears true for the SCIT task which has ITI of 100 ms. Wouldn't this artificially contaminate current trial data with the previous trial's data? What is meant by "pre-stimulus encoding periods"?
2. The two types of contextual cues at encoding are quite different, and the nature of the corresponding judgement made during encoding was different as well. Do the results differ between the two categories? In addition, since the configuration of the encoding task likely involves selective attention (i.e., attending to scene or color based on the judgement), do the results differ for subsequent context judgement made on the attended vs. unattended contexts?
3. Were the ocular artifacts identified via correlation with the HEOG and VEOG signals or with some other criteria?
4. Typos. Line 585: the average bar should not cover (1-alpha); Line 301: "[...] that that [...]"

In sum: The present work has a number of strengths and holds promise for having substantial impact. Addresses the above points will further bolster the evidence for the conclusions, and thus promises to further strengthen the work.

(Remarks on code availability)

Reviewer #2

(Remarks to the Author)

The authors sought to test the hypothesis that investigating trial-to-trial fluctuations in the levels of engagement of sustained attention, selective attention, and visual perception during study will improve the ability to predict which events would later be remembered vs. forgotten. Participants performed an episodic memory task as well as visual perception, sustained attention and selective attention tasks while scalp EEG was recorded. Using a "transfer learning" machine learning approach, the authors leveraged the engagement of the brain signals associated with the three underlying cognitive functions to predict episodic memory performance via encoding-related brain signals. They find that incorporating information about visual perception, sustained attention, and selective attention during encoding improves prediction performance. The authors show that although the three cognitive functions are not independent from one another, they each explain unique variance in encoding-related neural activity. They also show that encoding history (i.e. whether the preceding event was a hit or miss) influences engagement of the brain signals associated with the underlying cognitive functions. The authors conclude that successful encoding of episodic memories depends upon engagement of multiple underlying cognitive functions as well as prior encoding history.

The reported findings are compelling and the improved classification performance via the transfer learning approach will be valuable to the field. I do have some questions regarding conceptual aspects of the work and in general, I find the methods a bit hard to understand as currently written.

1. The authors hypothesize that "an event is more likely to involve higher levels of the underlying cognitive processes when it is preceded and followed by higher levels of engagement from those processes" (page 5, lines 118-120) and provide evidence for this account. However, this runs counter to the neural fatigue/camatosis hypothesis (Tulving & Rosenbaum, 2006, in *Distinctiveness and Memory*; Lohnas et al., 2020, *Neuropsychologia*) whereby continued use of a mechanism depletes said mechanism. The authors should clarify the motivation for their hypothesis and discuss possible accounts for why their findings run counter to the neural fatigue/camatosis hypothesis.

2. The authors include a control analysis to rule out the explanation that simply having more data will improve performance. However, the use of noise seems like a rather weak control. Would any additional task data -- and from any task, not just perception/attention tasks -- improve performance? To what extent would prediction accuracy without transfer learning increase if the number of electrodes increased?

3. The methods are a bit hard to follow and require the reader to read the authors' prior work. As it stands, it would be very difficult to replicate the current work based on the information in this paper. For instance, are the authors using LASSO regression? I did not see this mentioned anywhere in the current paper. I understand from the authors' prior work that utilizing more features yields better performance, but the justification for using both voltage and power in the same classifier needs to be explained as the two will be correlated at lower frequencies unless the authors removed condition-specific ERPs prior to calculating power. Furthermore, in some instances it seems as if only voltage is used (e.g. "the average covariance matrices of the trials of each class where the trials are matrices of EEG voltages across time bins and electrodes" page 27, lines 580-581). If only voltage is used, this should be clarified. If not, this should be corrected. It is not clear why bands instead of individual frequencies were chosen or exactly how 150 and then 40 features were selected. It is not currently clear the extent to which feature selection was independent from the primary classification analysis (this may be the five fold validation, but it wasn't clear). The exact performance metrics for excluding the three participants should be reported (e.g. above/below some cutoff value) as should the wave number for the Morlet wavelet analysis.

4. The authors find evidence to suggest that perception and attention decrease with time-on-task, which is consistent with prior work. However, they also report no time-on-task effect on memory performance, which conflicts with the assumption, and other results suggesting, that perception and attention contribute to memory encoding. The authors state that participants may become more efficient (Line 319); however, this calls into question the association between perception/attention processes and encoding. This relates somewhat to point #2, namely that it may be any cognitive task enhances prediction performance and it is not perception and attention processes per se that are driving the effects (although conceptually this is intuitive).

5. Although the rationale for predicting subsequent memory is clear, it is not definitive that a subsequently forgotten item was not actually encoded. In theory this should only hurt the ability to classify the two item classes, though this may account for why prior work has found less robust classification performance. Some consideration of this point is warranted.

6. The work of Katherine Duncan (Duncan et al., 2012, *Science*; Patel & Duncan, 2018, *Psychological Science*) should be referenced when discussing lingering brain states (e.g. "brain states last beyond a single event" page 16, lines 322-323). The Lohnas paper is cited here, but that paper provides evidence in support of depletion of resources which is not consistent with the claims being made.

(Remarks on code availability)

Code is written in Matlab which requires a license for use.

Reviewer #3

(Remarks to the Author)

Overview:

Previous literature has shown that it is possible to decode successful compared to unsuccessful episodic memory encoding. But it is difficult to decode encoding accuracy using a unidimensional approach. The present paper examines whether

classification of encoding with EEG is improved by considering component processes (visual perception and attention). The authors used transfer learning, a form of pattern analysis, with visual perception, sustained attention, and selective attention as the sources and episodic memory as the target to compare multidimensional pattern analysis to unidimensional pattern analysis in predicting memory encoding accuracy. The results showed that multidimensional pattern analysis improved encoding prediction. In addition, these improvements were greater than simply adding noise and also showed that source activity decreased over time and carried over from trial to trial. The authors conclude that using the multidimensional approach of transfer learning is a promising approach not only to encoding prediction but could be used in multiple cognitive domains.

This is an interesting paper that addresses important questions about the neural correlates of memory encoding and using brain data to predict memory performance. The main concern I have with the paper is that the authors do not provide sufficient justification for some aspects of their analysis nor do they sufficiently discuss the importance of these results.

SPECIFIC COMMENTS

1. The introduction describes the cognitive components used as sources (visual perception and attention). But it was not clear why the authors chose these sources. What about other cognitive functions that contribute to encoding such as executive function?
2. The rationale behind some aspects of the data analysis was not clear. First, it was not clear why the authors used binary classification for the visual perception task, but a reaction time cutoff for classification for the attention tasks. Second, it was not clear why the reaction time cutoff was 40%. Third, it was not clear what the optimal classification performance or running time was.
3. The authors do not thoroughly describe what unique features of the sources contributed to improved prediction. Were specific sources or oscillatory frequencies contributing more to prediction accuracy at different timepoints?
4. Although the authors address all of the results in the discussion, the present paper does not sufficiently discuss the importance of these results. The novelty of the study is the multidimensional approach leading to improved prediction and possible interventions to improve memory and should be described more thoroughly in the discussion and the conclusion.

Comments by page:

(p. 13) "Time x Condition or Source x Condition" should be stated "Time x Source x Condition".

(p. 14) "as unidimensional process" should be stated "as a unidimensional process".

(p. 22) "it was an old" should be stated "it was old".

(p. 31) "min-block" should be stated "mini-block".

(Remarks on code availability)

Reviewer #4

(Remarks to the Author)

A strength of this study was to use transfer learning to predict subsequent memory from multiple cognitive components (visual perception, sustained attention, and selective attention). The approach, beautifully summarized in Figure 1, is innovative.

This is an exciting study that deserves further development. However, a main concern is that it's not clear specifically what each cognitive component contributes to episodic memory success due to computational, conceptual, and anatomical ambiguity.

The computational ambiguity is that each cognitive component is not associated with a meaningful weighting. It's odd that step-wise order matters more than the specific components, so it's not possible to determine which cognitive domain is more or less relevant. It's reasonable to suspect shared variance, but then the authors should try to estimate it to partial out domain-specific variance. Contrary to the title, the components add up to less than the sum of the parts.

Although it was useful to test for the effects of adding any external source, the use of random noise was a weak control. A more appropriate control would be to shuffle high and low performance assignments to the existing EEG data. The second control of excluding the memory data from training was good, but it seems to diminish the value of the sources as independent predictors. An even better control would be to demonstrate independence from a cognitive source less relevant to the memory encoding task here. Irrelevant shams could include a task involving response interference, decision making, or auditory perception.

A final limitation is the lack of any anatomical information. fMRI data could be insightful here. This may be beyond the scope of the present study, but currently the neuroscientific and cognitive insights are minimal. Hence, this may be more appropriate for a specialized journal focused on methods.

The Discussion section could be shorter and more pertinent. Most of the content restated the results and was redundant. It should focus more on the contributions to the literature and the study's limitations.

(Remarks on code availability)

Reviewer #5

(Remarks to the Author)

I co-reviewed this manuscript with one of the reviewers who provided the listed reports. This is part of the Nature Communications initiative to facilitate training in peer review and to provide appropriate recognition for Early Career Researchers who co-review manuscripts

(Remarks on code availability)

Version 1:

Reviewer comments:

Reviewer #1

(Remarks to the Author)

In this revised manuscript, the authors were highly responsive to reviewer input. In so doing, the manuscript was further strengthened. The work is likely to be of broad interest and impact.

(Remarks on code availability)

Reviewer #2

(Remarks to the Author)

Overall I found this revised manuscript substantially improved and I appreciate the work that the authors did to address previous comments. Most of my concerns have been addressed, I still have two remaining points.

1. It would be worth noting in the limitations section in the discussion that there are some inherent limitations in the transfer learning approach. Although the authors clearly demonstrate that this approach outperforms alternative approaches, the fact that the algorithm

- cannot operate across participants
- requires the source and target to have the same length of data to be able to transfer the source information to the target data
- cannot generate evidence scores for every single trial and appears to be impacted by the order of events (e.g. indices being different across participants)

are all likely to limit the potential utility of this method. There exist cross-task, cross-time, cross-participant classification approaches that do not have these limitations (and do not rely on absolute values of brain signals as is described in the methods, p 33) which may not outperform the current methods, but are substantially easier to implement.

2. The rationale for including both voltage and power is still a bit vague. The authors state that "extracting as much information as possible from the neural activity allows us to have stronger prediction power" and then reference the supplement, but the referenced section does not indicate that having voltage and power outperforms either feature type (e.g. power) alone. Depending on how a classifier approaches sparsity and overfitting, correlated features may not actually contribute anything unique and the weights ascribed to said features may be somewhat arbitrary. Given the correlation between low frequencies and voltage deflections, it is not evident that more information is gained by including essentially redundant features.

(Remarks on code availability)

Reviewer #3

(Remarks to the Author)

The authors have done a nice job addressing the reviewers concerns.

(Remarks on code availability)

Reviewer #4

(Remarks to the Author)

It's good to see different reviewers raise related concerns. The authors were very responsive and the manuscript is much improved. The clarification of regression weights, the new control analyses using randomly shuffled labels, and the documentation of individual differences address this reviewer's major concerns. In future extensions, it will be interesting to see how the methods may pick up changes in response to experimental manipulations of the different cognitive components (e.g., increasing or degrading perceptual salience, boosting sustained attention with reward or emotion, etc.).

(Remarks on code availability)

Reviewer #5

(Remarks to the Author)

(Remarks on code availability)

We'd like to genuinely thank all the reviewers for their constructive feedback to remarkably improve the study and its impact on the literature. The paper has been extensively revised to incorporate reviewers' comments and suggestions; we feel that these changes have significantly improved the manuscript. Here, we include our point-by-point response to the reviews (in green) for our initial submission to *Nature Communications*.

Reviewer #1 (Remarks to the Author):

Major Comments:

1. Feature-selection procedure and its effects on the results

a. The reported analyses demonstrate the impact of having additional predictors from other tasks. However, given the approach to feature selection, one might worry that this transfer occurs because potentially predictive features from the memory task were excluded during feature selection and thus the memory-to-memory prediction is an underestimate of what is achievable. That is, could the finding that direct classification in the memory task is lower than classification when also including across-task predictors reflect the arbitrarily (low) number of features used to predict performance? While the analytic pipeline begins with 150 features, they are then reduced to the top 5 features. Some of the 145 excluded features might be/likely are informative, and their inclusion might boost unidimensional performance to be close to that of the multidimensional approach. To address this question, it would be helpful to report a plot of classification/prediction performance a function of the number of features included in the model, spanning from 5 up to the full 150 (using L2 or some other form of regularization to handle collinearity). Such a plot should be reported for both the unidimensional (memory-to-memory) and multidimensional approaches.

Thank you very much for your thoughtful comment. To ensure that the multidimensional improvement is not simply due to the feature selection approach, we performed the proposed analysis. Specifically, among the 150 extracted features, we performed the unidimensional and multidimensional classification by selecting the set of best 5, 10, 25, 50, and all 150 features. We found that selecting more features does indeed improve the unidimensional classification to an extent (until the classifier starts to overfit), but a similar pattern was found for the multidimensional classification. Moreover, even at the peak performance, the unidimensional classification did not reach the multidimensional classification performance even when 5 features were selected. We have now added these results to the manuscript. Specifically, on **Page 31**, we mentioned "Importantly, we selected the top 5 features (among the 150 extracted features) to ensure the classification performance would be high without sacrificing the running time. However, with the current feature selection methodology, it is possible that..."

Notably, these analyses are computationally time-consuming (e.g., selecting the top 50 features from 150 extracted features using the wrapper method require $\sim 6 \times 10^{104}$ classification training which takes several days to run) and we selected 10 random participants (as mentioned in the manuscript) to perform these analyses.

b. The results in the Supplementary Figure 9 do not speak against the above speculation. Assuming that those results were generated with the same feature-selection procedure, the fact that the top 5 features selected from the memory task did not add additional explanatory power to the selective attention classification while the reverse (attention-to-memory transfer) did increase classification suggests that the Top 5 features in the memory task do not overlap meaningfully with features that explain performance in the attention task. However, it remains a possibility that the Top 5 features in the other tasks are lower ranked features in the memory task and thus they were excluded from the latter; the "transfer learning" approach effectively adds them back in as additional predictors and performance increases. From this perspective, this improvement needn't depend on obtaining these

features from other tasks; rather, the features may well exist in the memory task itself and if these additional features from the memory task are included in the model (by including more than 5 features), then a similar performance increase might be obtained.

Thank you so much for this great point. We agree that without the analyses done to address comment 1a, it would be possible that the pattern found in the supplementary Fig. 13 is because of your suggested explanation. We hope the added analyses described above now sufficiently rule out this alternate possibility.

c. Since feature-selection was performed for each participant, can the authors comment on the extent of feature overlap across participants? If there is marked divergence across participants, given that some features (e.g., oscillation bands) have been tied to a variety of theoretical constructs, then it would be helpful to comment on what does it mean for different participants to have different sets of top features.

Thanks for your comment. It is worth mentioning that to extract CSP-based features, we first use spatial filters that project the data (i.e., voltage and power of different frequency bands over time) from our current electrodes to a new space so that the computed variance (within the time series of interest) is maximally different between high/low brain states. In other words, the projected electrodes in the new space are each a linear combination of our original electrodes. Given that these spatial filters will not be identical across participants, it is almost impossible to find the “exact” top feature overlap across participants.

Having said that, it is important to investigate the importance of different brain areas, time windows, sources, and frequency bands across the participants and we appreciate you for pointing this out. To briefly answer, there was a great amount of subject-to-subject variation in terms of how much each of the three sources were involved at different periods for hits and misses (Supplementary Fig. 10A and 10B). However, on average across all participants, no particular cognitive function at a certain encoding period consistently stood out as the most important factor for successful encoding. Moreover, while all frequency bands and voltage were important, the features extracted from gamma and then alpha frequency bands were most common to be selected as the top features. And in terms of the importance of the brain regions, the posterior areas for transferring perception, right areas for transferring sustained attention, and frontal areas for transferring selective attention information to the encoding data were most critical.

We have now addressed these points in full detail in the Supplementary information sections “Importance of different cognitive functions throughout the encoding period”, “Importance of different brain areas in the classification results”, and “Importance of different frequency bands and voltage in the classification results”. And we have referenced all of these sections in the main text.

d. A related conceptual question worth discussing is why “transfer learning” might be expected to improve performance over within-task training and testing. Specifically, in the “transfer learning” approach the ultimate evaluation is to test the out-of-task classifiers on predicting memory performance. For this transfer to occur, it must mean that the informative predictors from the other tasks are, themselves, present in the memory task data. Thus, shouldn't it be theoretically possible for a memory-task trained classifier (with more than 5 features) to learn from these informative features and thus lead to further performance boosts? What is it about out-of-task model training that leads to learning over these features that would not be possible from within-task model training?

That's a very good question and we appreciate that you brought this up. Briefly, no matter how many features we extract from the data, the associated label for each trial in the unidimensional approach will be simply 1 (hit) or 0 (miss). However, even for hits, some of the underlying processes at some encoding periods could potentially be at a *low* brain state (vice versa for the misses) and this is

something we can never find out when viewing memory as a unidimensional process. Thus, this multidimensional perspective allows us to better understand how high or low different underlying processes at different periods are, allowing us to better predict encoding success.

We have now addressed this point in the discussion. Specifically, on **page 18**, we have now mentioned “A conceptual question about transfer learning is why using the information from external tasks improves the prediction performance beyond what is achievable from within-task training and testing. To elaborate, to successfully...”.

2. As presented in the manuscript, both the rationale and the analyses conducted appear to rest on an implicit assumption that the three source tasks are process-pure, which is unlikely to be true and needs further comment. That is, there are likely multiple cognitive/neural processes that differentiate high and low trials in each of the tasks, including those framed by the authors but also additional processes. How do we know which of these processes for a given source task is producing the positive transfer effect? That is, what does the positive transfer tell us about the particular cognitive processes that differentiate successfully vs unsuccessfully encoded trials?

Thanks for your critical comment. We have now elaborated on this point in the discussion. Specifically, on **page 21**, we have mentioned: “First, it is unlikely that the three source tasks (or any other task) are completely “process-pure” and there are likely multiple cognitive processes that differentiate high and low trials in each of the tasks. For instance, earlier, we mentioned color perception and categorization as some of those potential subprocesses but there are likely additional subprocesses involved during some/all of the used tasks. Thus, *pinpointing* what exact subprocess for a given source task is producing the positive transfer effect (for each encoding event and for each participant) is beyond the scope of this study”.

3. Related to the previous point, the source tasks are minimally analyzed. It is unclear which components of each source task contribute to the classifier performance (e.g., is there a time-on-task or trial-history effect in some/all of the source tasks as well? For trial history effects, one could imagine that high/low performance trials could partly reflect differences in response priming/conflict, amongst other likely processing demands present in each task). If time-on-task and/or trial history effects are caused by some domain-general mechanisms independent of the specific cognitive processes you aim to measure, wouldn't they also affect the source tasks? How might this impact the logic of the transfer learning aspect of the work?

Thanks for your great comment! Briefly, the time-on-task effect was found on the sustained attention and selective attention tasks but not on the perception task (which was shorter than any other task we had). It is likely that the time-on-task effect is associated with domain-general mechanisms (including fatigue). This does not impact the logic of transfer learning as we have found that the sources are not fully independent from each other. We have addressed this point on **page 13**, “We repeated the same analysis for each source to inspect whether the time-on-task effect was driven by some domain-general processes (see Supplementary Information for the results, “The time-on-task effect for each source”). We found that the level of sustained...”

The history effect was only found on the sustained attention task though, and it is unlikely that it is associated with a domain-general mechanism that is shared across many cognitive tasks. In more detail, on **page 15**, we have stated “We repeated the same analysis for each source to inspect whether the history effect reflects some domain-general processes (see Supplementary Information for the results, “The history effect for each of the three sources”). We found that only during...”

4. The ability to transfer the features learned in the source tasks to the memory task relies on the

assumption that there is some degree of context-invariance in the measured cognitive processes. As a specific example, the neural correlates of selective attention would need to be similar in the selective-attention task and the memory task to observe “transfer learning”. Could unaccounted context-variance lead to an underestimate of the extent to which specific cognitive processes are involved in memory encoding? This point may be worth considering in the Discussion.

Thank you very much for pointing this out! We agree entirely. We have now mentioned this in the discussion. Specifically, on **page 17**, we have mentioned that “Importantly, the ability to transfer the features learned in a source task to the memory task relies on the implicit assumption that...”

5. Some details about the transfer-learning methods need further clarification:

- a. Please clarify why $D1$ is based only on the first row of w , which seems to be a column vector, so the first “row” of it is just one value. Does that mean the diagonal elements in $D1$ are all the same because other rows of w are not considered? The notation of “ $w1$ ” is also confusing because subscript has been used to denote class rather than row number up to this point.
- b. The description of C_i is also not clear. They “represent the average covariance matrices of the trials of each class where the trials are matrices of EEG voltages across time bins and electrodes”; what does “time bins” mean in this context? Were only raw voltages considered but not spectral features?
- c. Please specify the dimensionality of all variables in the equations, along with what those dimensions represent (e.g. is w a #channel-by-1 column vector?). Perhaps consider choosing different indexing methods when referring to different dimensions.

Thank you for your comments. We have now addressed all your questions on **pages 35 and 36** in the revision.

6. When framing the work in the Introduction, at times, some claims about the state of the literature appear overly strong or narrow. Specifically:

- a. On line 67, the claim that “no study has investigated the involvement of multiple cognitive functions during episodic encoding simultaneously” is a bit too strong. For example, multiple prior studies have examined the effects of various forms of attention on episodic encoding.

Thank you very much for your comment. We have now reframed that sentence to “However, despite episodic memory’s multidimensional nature, no study has considered the simultaneous involvement of multiple cognitive functions during episodic encoding to better understand the underlying reasons why any specific event was not successfully encoded.”

- b. Embedding the trial-history effect in the temporal-context framework seems unnecessarily narrow, as other mechanisms can produce similar effects (e.g. stimulus/response priming).

Thanks for pointing this out, Reviewer 2 also had a similar comment (comment #6). This is now addressed in both the introduction and discussion.

We also agree that one can speculate that response priming is potentially driving these effects. However, it is important to note that during retrieval, once the subject identifies an old item as old, they will need to make two additional context decisions about the color and the scene and only then, they are asked to make an old/new item for the next event, so it is unlikely that response priming is driving these effects. And during encoding, the responses are likely/unlikely context decisions which are not related to subsequently remembered/forgotten brain states. We have mentioned this in the discussion as well. Specifically, on **page 20**, we have mentioned: “Lastly, on a separate note, given that the encoding decisions are likely/unlikely context decisions and during the retrieval phase, an

item memory decision was followed by two context memory decisions before the next event's item memory decision, response priming cannot account for this result."

c. On line 77, the term "orthogonal tasks" would benefit from increased definition. Is the fact that the features transfer between tasks an apparent contradiction with the term "orthogonal"?

You are totally right, given that these source tasks are not fully independent from each other, the term "orthogonal" would be incorrect. We have now removed this term and reframed that part of the introduction. Thanks!

7. For trial-history effects (Fig 5), how were performance differences on the current trial controlled for? In other words, can one be sure that the effect is driven by performance on neighboring trials over and above the presumably different percentages of hits and misses in the current trial?

That's a great point! Please let us know if we understood your question correctly. Let's say this is the distribution of the events:

Events after a hit: 80% hits, 20% misses.

Events after a miss: 60% hits, 40% misses.

In that case, if we didn't control for different percentages of hits and misses, the results would be inaccurate. However, we have indeed averaged over hits and misses for the current trial. For example, the evidence scores for events after a hit are actually (the evidence scores for hits after a hit + the evidence scores for misses after a hit)/2.

This is mentioned in the methodology. Specifically, on **page 40**, we have mentioned "For each participant, we first averaged the evidence scores for each three sources for hits after a hit and averaged the evidence scores for each three sources for misses after a hit and then took an average to end up with average evidence scores of the three sources in three encoding periods for events after a hit. This was done similarly for events after a miss."

Minor Comments:

1. Perhaps this reflects a confusion, but did the EEG baseline (-400 to -600 ms) for episodic memory exceed the ITI for some trials (-350 to -700 ms)? The same (-200 to -400 ms) appears true for the SCIT task which has ITI of 100 ms. Wouldn't this artificially contaminate current trial data with the previous trial's data? What is meant by "pre-stimulus encoding periods"?

Thanks for bringing this up. You are right that for some encoding trials, the EEG baseline overlaps with the previous trial. This is actually not the case for the SCIT task though (as the epoch onset is when the cue is shown, not when the 1000-ms fixation screen is shown). This could be problematic, but since our CSP-based features are based on computing the variance, it does not matter what baseline period is chosen. In other words, no matter what the baseline period is, the variance-based features will remain the same.

And by "pre-stimulus encoding periods", we meant pre-stimulus periods for the encoding task. We have now corrected this in the manuscript.

2. The two types of contextual cues at encoding are quite different, and the nature of the corresponding judgement made during encoding was different as well. Do the results differ between the two categories? In addition, since the configuration of the encoding task likely involves selective attention (i.e., attending to scene or color based on the judgement), do the results differ for subsequent context judgement made on the attended vs. unattended contexts?

Thanks for your great comment! We have now repeated the transfer learning analyses for classifying attended context correct vs. incorrect decisions: 1) using only attend-color trials, and 2) using only attend-scene trials. Briefly, the context memory classification accuracy using the unidimensional approach was higher for attend-scene trials, but the attend-color trials benefitted more from the multidimensional approach. The results have now been added to the supplementary information for “The classification results to predict attended context memory success.”

Importantly, all the conducted analyses for context memory classification were performed on the attended context only (i.e., only the scene decision for the attend-scene and only the color decision for the attend-color trials mattered). We did not include the unattended context because the associated performances were near the chance level across participants.

3. Were the ocular artifacts identified via correlation with the HEOG and VEOG signals or with some other criteria?

As mentioned on **page 30**, “Components associated with eye movements were removed by visually inspecting the topographic component maps as well as the component time course”. Importantly, the HEOG and VEOG signals were included when ICA was run so the removed components would naturally be highly correlated with HEOG and VEOG signals.

4. Typos. Line 585: the average bar should not cover (1-alpha); Line 301: “[...] that that [...]”

Thank you, these have now been fixed.

In sum: The present work has a number of strengths and holds promise for having substantial impact. Addresses the above points will further bolster the evidence for the conclusions, and thus promises to further strengthen the work.

Thank you very much for your kind words and for your constructive feedback! We certainly agree that the suggested analyses and points were critical to improve the study and its impact.

Reviewer #2 (Remarks to the Author):

1. The authors hypothesize that "an event is more likely to involve higher levels of the underlying cognitive processes when it is preceded and followed by higher levels of engagement from those processes" (page 5, lines 118-120) and provide evidence for this account. However, this runs counter to the neural fatigue/camatosis hypothesis (Tulving & Rosenbaum, 2006, in *Distinctiveness and Memory*; Lohnas et al., 2020, *Neuropsychologia*) whereby continued use of a mechanism depletes said mechanism. The authors should clarify the motivation for their hypothesis and discuss possible accounts for why their findings run counter to the neural fatigue/camatosis hypothesis.

Thanks for your great comment! We totally agree that this point needs to be addressed. Briefly, in this analysis, we only considered the single prior event which does not allow us to test whether the event followed a *sustained* period of good encoding (in fact, the neural fatigue hypothesis is more in line with our time-on-task effect analysis). Moreover, given that the recognition performance in our task was better than the verbal recall performance Lohnas et al's study, the majority of the misses were after “good encoding history”, preventing us from replicating their study. In addition, opposite effects

were found for the hippocampus and the DLPFC, making it unclear what the scalp EEG data would look like.

We have now addressed this point in full detail in the discussion. Specifically, **on page 19**, it is noted “Our results are seemingly in contradiction to findings from Lohnas and colleagues who used intracranial EEG to show that non-recalled events with “good encoding history” (i.e., at least one of the two previous words was recalled) had worse encoding states based on the hippocampus activity than non-recalled events with “poor encoding history”...”

2. The authors include a control analysis to rule out the explanation that simply having more data will improve performance. However, the use of noise seems like a rather weak control. Would any additional task data -- and from any task, not just perception/attention tasks -- improve performance? To what extent would prediction accuracy without transfer learning increase if the number of electrodes increased?

Thank you so much for pointing this out! In fact, Reviewer 4 had a similar comment about the control analyses. They suggested shuffling high/low performance labels for the sources as a better control compared to adding noise. The analyses with the noise have now been replaced with the suggested control analysis to shuffle the source labels.

We also agree that including a “sham” source would be a very strong control analysis. Unfortunately, we did not collect any other neural data from the participants. Thus, we have now mentioned lacking the fourth, sham source as a limitation of the study. We have pointed this out in the discussion. **On page 22**, we now have added: “Although our control analyses support our assertion that the source processes (perception, selective, and sustained attention) were relevant to episodic encoding, in a future study, another good control would be to assess the contribution of neural activity from a task-irrelevant “sham” task (e.g. auditory perception).”

Regarding the impact of the number of electrodes, we agree that there could be some improvements if we had more electrodes. However, the previous studies that impacted the role of the number of EEG electrodes on the classification performance have not reported a high improvement from 32 to 64 electrodes:

<https://ieeexplore.ieee.org/abstract/document/6346274>

<https://ietresearch.onlinelibrary.wiley.com/doi/full/10.1049/joe.2018.9073>

In order to directly assess the impact of number of electrodes on transfer learning prediction accuracy, we repeated the analyses for 10 randomly selected participants on 16 of the 32 electrodes selected at random for each subject. The unidimensional classification performance decreased by 2.1% to 69.9%, and the multidimensional classification performance (i.e. transfer learning) decreased by 2.5% to 78.9%. Based on these results, we speculate that there would be a minor improvement in classification performance for both the unidimensional classification and multidimensional classification procedures. We have not currently included these findings in the manuscript but we're happy to add them if you think it would be helpful.

3. The methods are a bit hard to follow and require the reader to read the authors' prior work. As it stands, it would be very difficult to replicate the current work based on the information in this paper. For instance, are the authors using LASSO regression? I did not see this mentioned anywhere in the current paper. I understand from the authors' prior work that utilizing more features yields better performance, but the justification for using both voltage and power in the same classifier needs to be explained as the two will be correlated at lower frequencies unless the authors removed condition-specific ERPs prior to calculating power. Furthermore, in some instances it seems as if only voltage is used (e.g. "the average covariance matrices of the trials of each class where the trials are matrices of

EEG voltages across time bins and electrodes" page 27, lines 580-581). If only voltage is used, this should be clarified. If not, this should be corrected. It is not clear why bands instead of individual frequencies were chosen or exactly how 150 and then 40 features were selected. It is not currently clear the extent to which feature selection was independent from the primary classification analysis (this may be the five fold validation, but it wasn't clear). The exact performance metrics for excluding the three participants should be reported (e.g. above/below some cutoff value) as should the wave number for the Morlet wavelet analysis.

Thank you so much for pointing all these out! We have now addressed all these points in the manuscript, in the "Transferring the information from a source to the target" or "Participants" sections. We have also added a figure (supplementary Figure 7B) to make the procedure more clear and easier to replicate. Given that the changes are quite significant, it would be best if you could please see the changes in the "Transferring the information from a source to the target" section. In addition to that, in response to parts of your comment here:

- You are right that LASSO performs slightly better than naïve Bayes, but naïve Bayes is much faster than LASSO and that is why we chose naïve Bayes as our classifier. This is now mentioned in the manuscript.

- Both voltage and power were always used. This is now corrected, thank you for pointing this out.

- We could certainly use individual frequencies (which would lead to even more features) but in the EEG literature, it is more common to extract features from frequency bands since the neural activity associated with individual frequencies within a frequency band behave similarly to each other.

- It's correct that the feature selection is part of the *inner* five-fold cross-validation while the primary classification analysis is performed on the outer fold in our nested cross-validation procedure (supplementary Figure 6B).

- The only thing (in response to this comment) that is not discussed in the "Transferring the information from a source to the target" section is about the exclusion criterion. This is mentioned in the "Participants" section, on **page 23**, "We excluded 4 participants: 3 of them were outliers (more than 3 standard deviations below the mean) in terms of performance in at least 2 of the tasks..."

4. The authors find evidence to suggest that perception and attention decrease with time-on-task, which is consistent with prior work. However, they also report no time-on-task effect on memory performance, which conflicts with the assumption, and other results suggesting, that perception and attention contribute to memory encoding. The authors state that participants may become more efficient (Line 319); however, this calls into question the association between perception/attention processes and encoding. This relates somewhat to point #2, namely that it may be any cognitive task enhances prediction performance and it is not perception and attention processes per se that are driving the effects (although conceptually this is intuitive).

Thanks for your great point! Given that the memory performance does indeed decrease (even though it is not significant), it is plausible that both efficiency and neural fatigue are potentially contributing to the observed neural and behavioral effects. We have now addressed this point in full detail in the discussion. On **page 19**, we have now mentioned: "Regarding the time-on-task effect, we found the neural evidence for the underlying cognitive processes significantly decreased and while the memory performance also decreased over time, this decrease was not significant. Notably, previous neuroimaging studies..."

Overall, we agree that neural fatigue is impacting the results and the decrease in neural evidence for high levels of perception and attention engagement are consistent with this possibility.

We believe the analyses done in response to point #2 are supportive of the idea that the selected sources, and not just any cognitive source, are indeed relevant to the encoding processes.

5. Although the rationale for predicting subsequent memory is clear, it is not definitive that a subsequently forgotten item was not actually encoded. In theory this should only hurt the ability to classify the two item classes, though this may account for why prior work has found less robust classification performance. Some consideration of this point is warranted.

Really great point, thank you! We have now addressed this in the discussion. On **page 22**, we now have added: "The last limitation that is worth mentioning is related to all studies that attempt to predict encoding success, and this study is no exception. Specifically, it is not definitive that a subsequently forgotten event was not actually encoded. It could be the case that an event was weakly encoded and unsuccessfully consolidated. In addition, participants might make response errors during retrieval (i.e., responding "new" when they intended "old"), This means some "misses" might actually be associated with successful encoding, which could reduce the classification accuracy".

6. The work of Katherine Duncan (Duncan et al., 2012, Science; Patel & Duncan, 2018, Psychological Science) should be referenced when discussing lingering brain states (e.g. "brain states last beyond a single event" page 16, lines 322-323). The Lohnas paper is cited here, but that paper provides evidence in support of depletion of resources which is not consistent with the claims being made.

Thank you very much, Reviewer 1 also had a similar comment (comment #6b). This is now addressed on both the discussion and the introduction.

Reviewer #2 (Remarks on code availability):

Code is written in Matlab which requires a license for use.

Thanks for pointing this out. Based on our research, it seems for people who do not access University-based license, Octave can be a good alternative to run Matlab codes, and it does not require a license. We have now added this to the "READ ME FIRST.txt" document on the OSF's project directory.

Reviewer #3 (Remarks to the Author):

SPECIFIC COMMENTS

1. The introduction describes the cognitive components used as sources (visual perception and attention). But it was not clear why the authors chose these sources. What about other cognitive functions that contribute to encoding such as executive function?

Thank you very much for your great point! We have now justified our decision on the introduction and in the limitation section of the discussion, mentioned that future studies are needed to investigate the involvement of more related cognitive functions during episodic encoding.

Specifically, on **page 4**, we have mentioned: "Although there are several cognitive functions linked to episodic encoding, considering all of them in this study was not practical due to the length of the experiment session. We used visual perception, sustained attention, and selective attention as the sources as they have been closely linked to episodic encoding^{5–10} and their associated tasks are simple while allowing us to perform a high vs. low performance classifier."

Moreover, on **page 24**, we have mentioned: "In addition, we only investigated the engagement of processes associated with perception, sustained attention, and selective attention even though numerous cognitive functions..."

2. The rationale behind some aspects of the data analysis was not clear. First, it was not clear why the authors used binary classification for the visual perception task, but a reaction time cutoff for classification for the attention tasks. Second, it was not clear why the reaction time cutoff was 40%. Third, it was not clear what the optimal classification performance or running time was.

Thanks for pointing these out! The reason we didn't perform correct vs. incorrect classification for the attention tasks was that the performance for those two tasks was on the ceiling level. Please note that we *are* performing binary classifiers for both of those tasks. The fastest 40% trials get the "high" label while the slowest 40% trials get the "low" label. And we have now explained the rationale for the top/bottom 40% cutoff. For both of attention tasks' description, on **pages 25** and **26**, we have now mentioned: "The top 40% and bottom 40% cutoff ensured that we are including as many trials as possible while having a clear boundary between "high" and "low" trials. We did not classify correct vs. incorrect decisions since the performance was on the ceiling level."

And thanks for your comment regarding the optimal performance and the running time. In fact, Reviewer's 1 comment 1 is kind of related to this point too! We have now added Supplementary Fig. 5 which shows how much the classification performance and running time change based on 1- the number of features the wrapper searches through and 2- the number of features selected for classification. Based on that figure that has now been added, we found that filtering the top 40 features and selecting the best 5 features among them with the wrapper method ensured the optimal classification performance with reasonable running time

3. The authors do not thoroughly describe what unique features of the sources contributed to improved prediction. Were specific sources or oscillatory frequencies contributing more to prediction accuracy at different timepoints?

Thank you very much for bring up such an important point! In fact, Reviewer 4 and Reviewer 1 (in their 1C comment) similar comments about getting more insight into the classification results. To briefly answer, there was a great amount of subject-to-subject variation in terms of how much each of the three sources were involved at different periods for hits and misses (Supplementary Fig. 10A and 10B). However, on average across all participants, no particular cognitive function at a certain encoding period consistently stood out as the most important factor for successful encoding. Moreover, while all frequency bands and voltage were important, the features extracted from gamma and then alpha frequency bands were most common to be selected as the top features.

We have now addressed these points in full detail in the Supplementary information sections "Importance of different cognitive functions throughout the encoding period" and "Importance of different frequency bands and voltage in the classification results". And we have referenced all of these sections in the main text.

4. Although the authors address all of the results in the discussion, the present paper does not sufficiently discuss the importance of these results. The novelty of the study is the multidimensional approach leading to improved prediction and possible interventions to improve memory and should be described more thoroughly in the discussion and the conclusion.

Thanks for your comment. In fact, Reviewer #4 had a similar comment regarding the discussion, and we made important changes to the discussion. We have now significantly cut the parts where the results are being repeated and we now discuss the implications of this study and the limitation in more detail. For example, to emphasize the novelty of this study, on **page 22**, we have mentioned: "Critically, not only does this study shed light on the multidimensionality of memory, which is

important for basic science, but it also opens avenues for future implications in terms of real-world interventions to improve memory. This multidimensional evaluation approach substantially improved prediction accuracy but more importantly, perhaps, it offers potential to personalize future feedback systems that could be implemented for real-world intervention applications. To elaborate,...

Comments by page:

(p. 13) "Time x Condition or Source x Condition" should be stated "Time x Source x Condition".
Thanks, this is now corrected.

(p. 14) "as unidimensional process" should be stated "as a unidimensional process".
We have now fixed it, thank you.

(p. 22) "it was an old" should be stated "it was old".
Thanks, this is now addressed.

(p. 31) "min-block" should be stated "mini-block".
This is now addressed. Thanks for pointing all these out!

Reviewer #4 (Remarks to the Author):

The computational ambiguity is that each cognitive component is not associated with a meaningful weighting. It's odd that step-wise order matters more than the specific components, so it's not possible to determine which cognitive domain is more or less relevant. It's reasonable to suspect shared variance, but then the authors should try to estimate it to partial out domain-specific variance. Contrary to the title, the components add up to less than the sum of the parts.

Thank you very much for your great comment! In fact, Reviewer 3 (in their 3rd comment) had a similar point about the importance of different cognitive functions. Briefly speaking, the regression weights give us critical insight into the importance of different cognitive functions at different encoding periods for each specific subject. So, each cognitive component *is* associated with a meaningful weighting, and we should have mentioned this in our initial submission, as you perfectly pointed it out. We have now pointed this out on **page 38**, "This technique associates larger regression coefficients with sources and periods that are particularly important for determining memory success and smaller coefficients to the sources and periods that are not as important for determining memory success. As a result, the linear regression coefficients provide critical insight into which cognitive functions during which encoding periods are particularly essential for memory formation for a participant (see Supplementary Fig. 10A and 10B).".

Importantly, there was a great amount of *subject-to-subject variation* in terms of how much each of the three sources were involved at different periods for hits and misses (Supplementary Fig. 10A and 10B). However, on average across all participants, no particular cognitive function at a certain encoding period consistently stood out as the most important factor for successful encoding. We have now addressed this point in full detail in the Supplementary information section "Importance of different cognitive functions throughout the encoding period". And we have referenced this on the main text, on **page 11**, "Specifically, is one of visual perception, sustained attention, and selective attention more important than the other two cognitive functions at different encoding periods for successful memory formation? As seen in Supplementary Fig. 10C, across all participants, there was no cognitive function at a particular encoding period that..."

Although it was useful to test for the effects of adding any external source, the use of random noise was a weak control. A more appropriate control would be to shuffle high and low performance assignments to the existing EEG data. The second control of excluding the memory data from training was good, but it seems to diminish the value of the sources as independent predictors. An even better control would be to demonstrate independence from a cognitive source less relevant to the memory encoding task here. Irrelevant shams could include a task involving response interference, decision making, or auditory perception.

Thank you so much for suggesting these great control analyses! In fact, Reviewer 2, in their second comment, had a similar point. We have now replaced the noise control analysis with the high-low source performance shuffling analysis.

We agree that including a sham source would be a very strong control analysis. Unfortunately, we did not collect any other neural data from the participants. Thus, we have now mentioned lacking the fourth, sham source as a limitation of the study. We have pointed this out in the discussion. On **page 22**, we now have added: “Although our control analyses support our assertion that the source processes (perception, selective, and sustained attention) were relevant to episodic encoding, in a future study, another good control would be to assess the contribution of neural activity from a task-irrelevant “sham” task (e.g. auditory perception).”.

A final limitation is the lack of any anatomical information. fMRI data could be insightful here. This may be beyond the scope of the present study, but currently the neuroscientific and cognitive insights are minimal. Hence, this may be more appropriate for a specialized journal focused on methods.

It is true that EEG does not have high spatial resolution compared to fMRI, but the topography information could provide an insight into which parts of the brain contributed the most when transferring a source to the target. Specifically, the posterior areas received the highest importance when transferring the perception information to the encoding data. Moreover, there was a right lateralization when transferring the sustained attention information to the encoding activity. Lastly, the frontal regions were most critical when transferring the selective attention information to the encoding activity. We have now discussed these in the “Importance of different brain areas in the classification results” section in the supplementary information.

We have also pointed this out in the discussion. Specifically, on **page 22**, we have mentioned “Furthermore, using functional magnetic resonance imaging (fMRI) could better elucidate the brain networks and areas that are involved when assessing the engagement of different cognitive functions during encoding. However, assessing the spatial information regarding the selected features was still informative (see Supplementary information “Importance of different brain areas in the classification results”).”

The Discussion section could be shorter and more pertinent. Most of the content restated the results and was redundant. It should focus more on the contributions to the literature and the study’s limitations.

Thanks for your very important comment! In fact, Reviewer #3 had a similar comment regarding the discussion (comment 4), and we made important changes to the discussion. We have now significantly cut the parts where the results are being repeated and we now discuss the implications of this study and the limitations in more detail.

Reviewer #5 (Remarks to the Author):

I co-reviewed this manuscript with one of the reviewers who provided the listed reports. This is part of

the Nature Communications initiative to facilitate training in peer review and to provide appropriate recognition for Early Career Researchers who co-review manuscripts

Here, we include our point-by-point response to the reviews (in green) for our revised submission to *Nature Communications*.

Reviewer #2:

1. It would be worth noting in the limitations section in the discussion that there are some inherent limitations in the transfer learning approach. Although the authors clearly demonstrate that this approach outperforms alternative approaches, the fact that the algorithm
 - cannot operate across participants
 - requires the source and target to have the same length of data to be able to transfer the source information to the target data
 - cannot generate evidence scores for every single trial and appears to be impacted by the order of events (e.g. indices being different across participants)

are all likely to limit the potential utility of this method. There exist cross-task, cross-time, cross-participant classification approaches that do not have these limitations (and do not rely on absolute values of brain signals as is described in the methods, p 33) which may not outperform the current methods, but are substantially easier to implement.

Thank you very much for your thoughtful comment. Regarding the first point, we should clarify that transfer learning can indeed be used to transfer the information from one participant to another participant. In fact, this is the primary application of the transfer learning algorithms, as mentioned on **page 4**: “Notably, only a handful of EEG studies have used transfer learning with multiple sources with the sources typically being EEG activity associated with the same cognitive task but from different participants with the aim of designing a subject-independent classifier that could potentially be useful for a brain-computer interface (BCI) system”. We have now made this clearer in the method section. Specifically, on **page 37**, we have added “It is critical to note that the same principles would apply to transfer the information from one participant to another. In that case, the source would be the information from the “source subject” and the target would be the information (from the same or even a different task) from the “target subject”. Since we did not have any specific hypothesis regarding between-subject classification, we did not perform any cross-subject classification analysis.”

Regarding the point on the source and target length, we have now made this clear, on **page 38**, that the number of *time-points* between the source and the target has to be the same (since the same variance-based features will be extracted from both source and the target). On **Page 38**, we have also added: “Having said that, if one wants to transfer a longer period of source to a shorter period of the target, they can use moving averaging of the data to reduce the number of time samples of the source and then perform the transfer learning analyses.”

Regarding the last point, we’re not fully sure what the reviewer meant. It is important to note that every single trial will receive a temporal map of evidence scores from each 3 sources (Supplementary Fig. 7). By indices being different, we think the reviewer is pointing out to the time-on-task effect analyses. Given that not every trial will be a hit (or a miss), and given that some trials were excluded because of noise, we extrapolated the results from the available indices to measure the extent to which the evidence scores change from the 1st mini-block to the 16th mini-block. It is correct that not every subject will have the same indices for their hits or misses (i.e., subject 1 has trials 22, 48, 103, ..., and 201 as misses while subject 2 has other trials as misses), but this limitation is not related to the transfer learning algorithm. On **Page 40**, we have now mentioned: “To inspect how the evidence scores for the three sources change from the 1st event to the 240th event for hits and misses separately, it would be ideal to have the evidence scores for every single trial number for both hits and misses for every participant. But this is simply not possible and is a limitation of this analysis.”

Lastly, the algorithm does not rely on “absolute values of brain signals”. As mentioned in the manuscript, the example mention on page 34 is simply “a simple hypothetical example” to give an intuition of what it means to transfer the perception information to the encoding data. To make this clearer, we have now added “and this is not reflective of how the proposed algorithm exactly works,” on **page 34**.

2. The rationale for including both voltage and power is still a bit vague. The authors state that "extracting as much information as possible from the neural activity allows us to have stronger prediction power" and then reference the supplement, but the referenced section does not indicate that having voltage and power outperforms either feature type (e.g. power) alone. Depending on how a classifier approaches sparsity and overfitting, correlated features may not actually contribute anything unique and the weights ascribed to said features may be somewhat arbitrary. Given the correlation between low frequencies and voltage deflections, it is not evident that more information is gained by including essentially redundant features.

Thanks for your comment. In the prior work that is cited in the manuscript, we established that extracting features from both voltage and power leads to better performances (across multiple tasks and modalities), on average across participants. Regarding your concern about correlation between voltage and low-frequency power, we want to note a few points. First, the power of 6-8 Hz was used in this study for theta, and we believe excluding 4-6 Hz helps with the concerns about correlation between the voltage and low-frequency power. More importantly, the critical role of the feature selection step used in this study is to avoid using redundant features that would not contribute uniquely and positively to the classification. Thus, even if there are features that are correlated in the pool of extracted features, they will be filtered out once feature selection is performed. Lastly, the nested-cross validation approach ensures that the test data was never used during the classifier's training, ensuring the classifier avoids overfitting.